



# Sources of Fe-binding organic ligands in surface waters of the western Antarctic Peninsula

Indah Ardiningsih[1], Kyyas Seyitmuhammedov[2], Sylvia G. Sander[3], Claudine H. Stirling[2], Gert-Jan Reichart[1,4], Kevin R. Arrigo[5], Loes J. A. Gerringa[1], Rob Middag[1,2]

[1]Royal Netherlands Institute for Sea Research (NIOZ), PO Box 59, 1790 AB Den Burg, Department of Ocean Systems, University of Utrecht, the Netherlands
[2]Centre for Trace Element Analysis and Chemistry Department, University of Otago, Dunedin, New Zealand
[3]International Atomic Energy Agency, 4a Quai Antoine 1er, 98000, Principality of Monaco, Monaco
[4]Earth and Geoscience Department, University of Utrecht, Utrecht, the Netherlands
[5]Department of Earth System Science, Stanford University, USA

*Correspondence to*: Indah Ardiningsih (Indah.ardiningsih@nioz.nl)

**Abstract.** Organic ligands are a key factor determining the availability of dissolved iron (DFe) in the high nutrient low chlorophyll (HNLC) areas of the Southern Ocean. In this study, organic speciation of Fe is investigated along a natural gradient of the western Antarctic Peninsula, from an ice covered shelf to the open ocean. An

electrochemical approach, competitive ligand exchange - adsorptive cathodic stripping voltammetry (CLE-AdCSV) was applied. Our results indicated that organic ligands in surface water on the shelf are associated with ice-algal exudates, possibly combined with melting of sea-ice. Organic ligands in deeper shelf water are supplied via resuspension of slope or shelf sediments. Further offshore, organic ligands are most likely related to the development of phytoplankton blooms in open ocean waters. On the shelf, total ligand concentrations ($[L_t]$) were

between 1.2 nM eq. Fe and 6.4 nM eq. Fe. The organic ligands offshore ranged between 1.0 and 3.0 nM eq. Fe. The southern boundary of the Antarctic Circumpolar Current (SB ACC) separated the organic ligands on the shelf from bloom-associated ligands offshore. Overall, organic ligand concentrations always exceeded DFe concentration (excess ligand concentration, $[L'] = 0.8 - 5.0$ nM eq. Fe). The $[L']$ made up to 80% of $[L_t]$, suggesting that any additional Fe input can be stabilized in the dissolved form via organic complexation. The

denser modified Circumpolar Deep Water (mCDW) on the shelf showed the highest complexation capacity of Fe ($\alpha_{Fe'L}$; the product of $[L']$ and conditional binding strength of ligands, $K_{Fe'L}^{cond}$). Since Fe is also supplied by shelf sediments and glacial discharge, the high complexation capacity over the shelf can keep Fe dissolved and available for local primary productivity later in the season, upon sea ice melting.



## 1. Introduction

The Southern Ocean is a High Nutrient Low Chlorophyll (HNLC; e.g. Sunda et al., 1989) region where the phytoplankton biomass is relatively low despite high ambient macronutrient concentrations, i.e. nitrogen (N), phosphorus (P) and silicon (Si) (e.g. Martin et al., 1991; Schoffman et al., 2016). The generally limited availability of light and the micronutrient iron (Fe) prevents phytoplankton from depleting P and N in the vast majority of HNLC areas (de Baar et al., 2005; de Baar, 1990; Martin et al., 1991; Viljoen et al., 2018). Indeed, Fe regulates

the dynamics of primary production as it is involved in various cellular processes (Schoffman et al., 2016; Sunda, 1989). In the HNLC Southern Ocean, the availability of Fe has a direct impact on the early spring phytoplankton bloom, and thus on primary productivity (Moore et al., 2013). The Fe limitation in the Southern Ocean could thus have a direct effect on the amount of atmospheric CO2 sequestration (Le Quéré et al., 2016; Arrigo et al., 2008; Raven and Falkowski, 1999) to the deep ocean via the biological pump (De La Rocha, 2006; Lam et al., 2011).

Accordingly, the availability of Fe in the Southern Ocean is not only important for sustaining the food web, but also has a substantial impact on global climate (Hanley et al., 2019 and references therein).

The low solubility of Fe in seawater, coupled with low atmospheric and terrestrial input of Fe, result in the scarcity of dissolved-Fe (DFe) in the Southern Ocean. In oxygenated seawater, Fe is mainly present in its oxidized form, Fe(III), predominantly as Fe(III)oxy-hydroxide species. These species tend to undergo further hydrolysis (Liu and

Millero, 2002) and are thereby removed from the water column by scavenging or precipitation processes. Organic Fe-binding ligands greatly elevate Fe solubility in seawater (Kuma et al., 1996) by stabilizing Fe in Fe-ligand complexes, and thus allowing Fe to remain longer in the water column. Moreover, Fe bound to organic ligands appears to be bioavailable to marine phytoplankton (Maldonado et al., 2005; Rijkenberg et al., 2008; Hassler et al., 2020). As such, organic ligands are a key component of Fe chemistry and bioavailability, notably in HNLC

regions, as illustrated by Lauderdale et al. (2020). These authors showed, with an idealized biogeochemical model of the ocean, that the interaction between microbial ligand production and binding of Fe by these ligands functions as a positive feedback to maintain the DFe standing stock in the oceans.

Various ligand groups exist and are classified based on their origin. Fe binding ligand properties can be measured by the competition against well-characterized artificial ligands with known stability constants. Analysis is done

using an electrochemical technique, competitive ligand exchange (CLE) - adsorptive cathodic stripping voltammetry (AdCSV). The application of AdCSV gives the total concentration ($[L_t]$) and conditional binding strength ($K_{Fe'L}^{cond}$) of the dissolved organic ligands but does not provide information on the identity of ligands.





Although the sources and identities of Fe-binding ligands are still largely unknown, these ligands have a biological origin, being either actively produced or passively generated through microbial activity.

Laboratory studies have documented the active production of Fe-binging ligands under Fe-limited conditions (Boiteau et al., 2013; Boiteau et al., 2016; Butler, 2005). Several types of siderophores, low-molecular-weight organic compounds which have strong affinity to Fe, are produced by mixed marine bacteria communities under Fe stress (Butler, 2005), suggesting that high ligand concentrations are related to a mechanism of Fe acquisition in an Fe-limited environment. These compounds have also been extracted (Boiteau et al., 2016; Macrellis et al.,

2001; Velasquez et al., 2011) or identified (Mawji et al., 2008; Velasquez et al., 2016) in field samples. However, they generally occur at picomolar levels (Boiteau et al., 2019) and are a small contributor to the total ligand pool. Other ligand types, such as polysaccharide compounds, are passively generated in situ from microbial excretion and grazing (Sato et al., 2007; Laglera et al., 2019b). The polysaccharides, such as exo-polymeric substances (EPS), are excreted abundantly by a large number of microbial cells, especially in surface water covered by with

sea-ice (Norman et al., 2015; Lannuzel et al., 2015). Although EPS are relatively labile macromolecules, they can be present in up to micromolar concentrations in seawater, showing the potential to outcompete stronger binding siderophores (Hassler et al., 2017). In addition, humic substances (HS) or HS-like substances from various origins constitute another type of ligand (Krachler et al., 2015; Laglera et al., 2019a; Whitby et al., 2020). Typically, HS are derived from remineralization and degradation of organic matter (Burkhardt et al., 2014). Terrestrial input of

organic matter can supply HS to estuarine and coastal areas, whereas sediment resuspension and upwelling often supply HS-like substances to the continental shelf (Gerringa et al., 2008; Buck et al., 2017). HS-like substances have also shown to be part of Fe binding ligands in biologically refractory deep ocean dissolved organic matter (rDOM) with low Fe-bioavailability. However, photodegradation of rDOM was shown to increase the Fe bioavailability making Fe bound to such substances an important source in HNLC areas where upwelling plays a

role (Hassler et al., 2020; Lauderdale et al., 2020; Whitby et al., 2020; Laglera et al., 2019a).

The Bellingshausen Sea along the western Antarctic Peninsula (WAP) region has a distinct natural DFe gradient (Sherrell et al., 2018; Moffat and Meredith, 2018). The hydrography is strongly influenced by the dynamics of shelf-ocean water exchange. The shoreward intrusion of Circumpolar Deep Water (CDW) provides macronutrients to the shelf region, whereas offshore-flowing waters supply the micronutrients Fe and Mn to the

open ocean from local sources (De Jong et al., 2015; Sherrell et al., 2018), such as glacial meltwater, sediments and upwelling. The shelf sea of the WAP is a biologically-rich marine ecosystem in the Southern Ocean. The abundance, community composition and trophic structure of marine primary producers are directly impacted by the changing ice conditions and longer periods of open water due to climate change (Turner et al., 2013).



Moreover, rapid increases in anthropogenic $CO_2$ has enhanced the air-sea $CO_2$ fluxes, decreasing the bulk seawater

pH, resulting in ocean acidification (Mikaloff Fletcher et al., 2006), which alters the physicochemical properties

of seawater and impacts the organic complexation of Fe (Ye et al., 2020). As the WAP has undergone significant

warming (Turner et al., 2020), the changes in ice conditions will influence the supply of Fe and organic ligands,

shaping the net primary production in this region. Understanding the sources and distribution of organic ligands

provides important information on DFe availability, which is a fundamental step towards understanding the impact

of warming of the Antarctic region on primary productivity in the Southern Ocean.

In this study, surface waters were sampled in a region of mixing between shelf-influenced waters and HNLC

waters in the Bellingshausen Sea along the WAP. In order to probe sources and distributions of Fe-binding ligands

along a natural gradient of Fe, the CLE-AdCSV technique was used to quantify the total concentrations and

conditional stability constants of Fe-binding ligands.

## 2.   Material and methods

### 2.1  Sample collection

Samples were collected onboard the research vessel Nathaniel B. Palmer (Cruise NBP1409) during the austral

spring between 31 October and 21 November 2014 in the Bellingshausen Sea, west of the Antarctic Peninsula

(Figure 1). Water samples were obtained from the surface to maximum 600 m depth at five stations along a

transect extending from Adelaide Island on the shelf out into the Bellingshausen Sea in a northwest direction

(Figure 1). St. 70 and 72 are situated near the shore in the shelf sea, St. 96 is located at the continental shelf break,

and St. 84 and 90 are located offshore over deep waters.

Hydrographic parameters were measured using a conventional conductivity-temperature-depth (CTD) rosette

equipped with a fluorometer (WET Labs ECOAFL/FL) and an oxygen sensor (SBE-43). Seawater samples for

DFe and Fe-binding ligands in this study were obtained using GO-FLO bottles attached to a Kevlar® wire.

Seawater samples were filtered over 0.2 µm filters (Satroban 300, Sartorius®) into pre-cleaned sample bottles

inside a trace metal clean van. Sample bottles were pre-cleaned following a three-step cleaning protocol for trace

element sample bottles (Middag et al., 2009).

Filtered seawater samples for Fe-binding ligand analysis were collected into acid-cleaned 500 mL low density

polyethylene (LDPE, Nalgene) bottles and stored at -20ºC immediately after collection. For DFe analysis, filtered

seawater was collected into acid-cleaned 60 mL LDPE (Nalgene), and acidified to a pH of 1.8 with concentrated





quartz-distilled hydrochloric acid (HCl) to give a final concentration of 0.024 M HCl. Filtered samples for macronutrient concentration analysis were stored at -20ºC (for N samples) and at 4ºC (for Si samples).

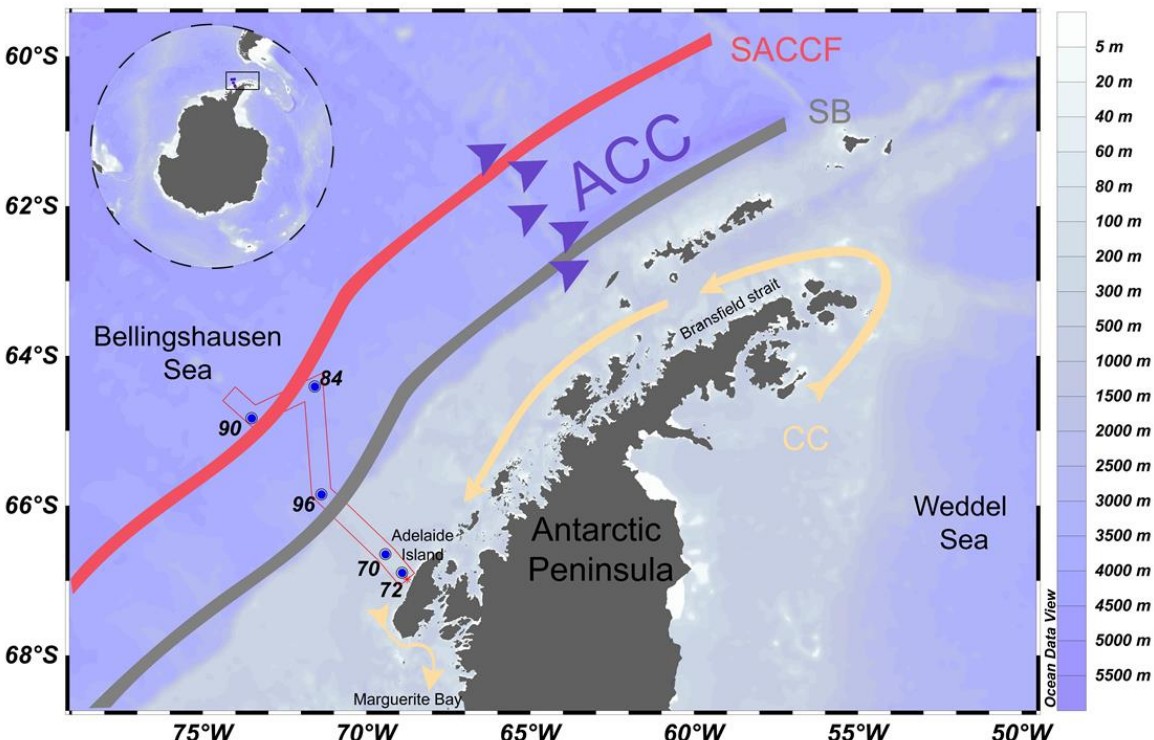

**Figure 1: Map of the sampling sites along our study transect near the Western Antarctic Peninsula. The stations are indicated by blue dots and station numbers. The Antarctic Circumpolar Current (ACC) is indicated by purple arrows. The Coastal Current (CC) is indicated by a yellow arrows. The Southern Boundary (SB) of ACC Front is indicated by the grey line. The southern ACC front (SACCF) is indicated by the red line.**

### 2.2 Analysis of DFe and nutrients

The DFe analysis is described in detail by Seyitmuhammedov (2020). In short the DFe analysis was conducted using high-resolution inductively coupled plasma mass spectrometry (HR-ICP-MS) using a Thermo Fisher Element XR instrument at NIOZ, the Netherlands and with additional inter-calibration using an Amtek Nu Attom instrument at University of Otago, New Zealand, after preconcentration using an automated seaFAST system (SC-4 DX seaFAST pico; ESI). The quantification was done via standard additions. Accuracy and reproducibility were

monitored by regular measurements of the reference materials SAFe D1 and GEOTRACES South Pacific (GSP) seawater, and an in-house reference seawater sample, North Atlantic Deep Water (NADW). Results for DFe



analyses of reference materials were within the range of 0.722 ± 0.008 nM (n = 3) for SAFe D1 2013 and 0.155 ± 0.045 nM (n = 13) for GSP 2019 consensus values. The average overall method blank (seaFAST and ICP-MS) concentration, determined by measuring acidified ultrapure water as a sample, was 0.05 ± 0.02 nM (n = 21).

Macronutrients (N and Si) were analyzed simultaneously with a discrete autoanalyzer TRAACS 800 (Technicon) in the shore-based laboratory at NIOZ.

### 2.3  Analysis of Fe-binding ligands

Samples were thawed in the dark and vigorously shaken prior to further treatment. Electrochemical analysis CLE-AdCSV with salicylaldoxime (SA) as a competing added ligand (Abualhaija and van den Berg, 2014) was used.

In short, the voltammetric system consisted of a BioAnalytical System (BASi) controlled growth mercury electrode connected to an Epsilon 2 analyzer (BASi). The voltammetric system was controlled using ECDsoft interface software. The electrodes in the voltammetric stand included a standard Hg drop working electrode, a platinum wire counter electrode, and a double-junction Ag/AgCl reference electrode (3M KCl).

For the titration, 10 mL sample aliquots were added to 12 pre-conditioned Teflon (Fluorinated Ethylene Propylene

(FEP), Savillex) vials and buffered to seawater a pH of 8.2 with 0.1 M ammonium-borate buffer. The sample aliquots were titrated with Fe from 0 to 10 nM (with a 0.5 nM interval from 0 to 3 nM; and with a 2 nM interval from 4 to 10 nM Fe) and vials without Fe addition were prepared twice. Then, the competing ligand, SA, was added at a final concentration of 5 µM. The mixture was left to equilibrate for at least 8 hours or typically overnight (Abualhaija and van den Berg, 2014). Before analysis, the Teflon vials for titration were pre-conditioned at least

three times with seawater containing SA and the intended Fe addition. For each titration point, duplicate scans were done in the same Teflon vial as voltammetric cell.

### 2.4  Calculation of Fe speciation

Ligand parameters, $[L_t]$ and $K_{Fe'L}^{cond}$, were obtained by fitting the data from the CLE-CSV titration into a non-linear Langmuir model. One- and a two-ligand models were applied, assuming one ligand and two ligand groups existed,

respectively. The R software package was used for data fitting (Gerringa et al., 2014). The $[L_t]$ reported in nM eq. Fe and the conditional stability constant values are reported as log $K_{Fe'L}^{cond}$.

The values of $[L_t]$, log $K_{Fe'L}^{cond}$, and DFe were used to calculate the concentration of natural unbound ligands, the excess ligands concentration $[L']$, and the side reaction coefficient of ligands ($\alpha_{Fe'L}$; the product of $[L']$ and log $K_{Fe'L}^{cond}$, Gledhill and Gerringa (2017). The prime symbol ($'$) in excess ligand concentrations denotes the free ligands



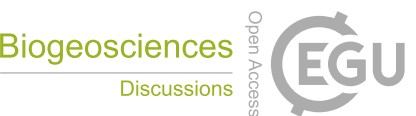

not bound to Fe, whereas the prime symbol after Fe denotes the inorganic fraction of Fe. The value of $\alpha_{FeL}$ is

presented as a logarithmic value (log $\alpha_{Fe'L}$) and referred to as complexation capacity.

The value of the inorganic Fe side reaction coefficient ($\alpha_{Fe'}$) was determined using the hydrolysis constants of Liu

and Millero (1999) at S=36 and at pH of the analysis (pH=8.2). Hence, the value of log $\alpha_{Fe\text{-inorganic}}$ =10.4 was used

in the calculation for Fe speciation. The conditional stability constant of SA (log $K_{Fe'(SA)}^{cond}$ = 6.76) used in this study

is based on the calibration of SA against diethylenetriaminepentaacetic acid by Gerringa et al., (submitted).

## 3.  Results

### 3.1 Hydrography

Water masses were identified by plotting the Conservative Temperature (Θ) versus the Absolute Salinity (S$_A$)
(Tomczak and Godfrey, 2003) as generated by the freeware ODV (Schlitzer, 2018) from CTD data (Figure 2a).

The water mass description follows the definitions of Klinck et al. (2004) and Smith et al. (1999). A detailed
description of hydrographic features of the WAP is described elsewhere (Moffat and Meredith, 2018; Klinck et
al., 2004; Smith et al., 1999) and briefly summarized here.

Two distinct horizontal currents exist in the study area, the Coastal Current (CC) and the Antarctic Circumpolar
Current (ACC) (Figure 1). In the vicinity of the WAP, the ACC is a large strong eastward flowing current

bordering the outer continental shelf. The CC is a strong but narrow southwesterly flowing current that is forced
by freshwater discharge and wind over the shelf (Grotov et al., 1998; Moffat and Meredith, 2018). The CC flows
along the coast of the WAP from Bransfield Strait to Marguerite Bay (Figure 1). Our sampling stations are located
along a transect that is perpendicular to the prevailing currents. A number of oceanographic fronts exist along the
ACC (Orsi et al., 1995). Two fronts intersect our sampling transect, namely the southern ACC Front and the

southern boundary (SB) of the ACC. These two fronts are present oceanward from the shelf (Figure 1), usually
about 50 – 200 km apart. Although their positions vary, these fronts remain approximately parallel with the
continental shelf break along the WAP (Moffat and Meredith, 2018).





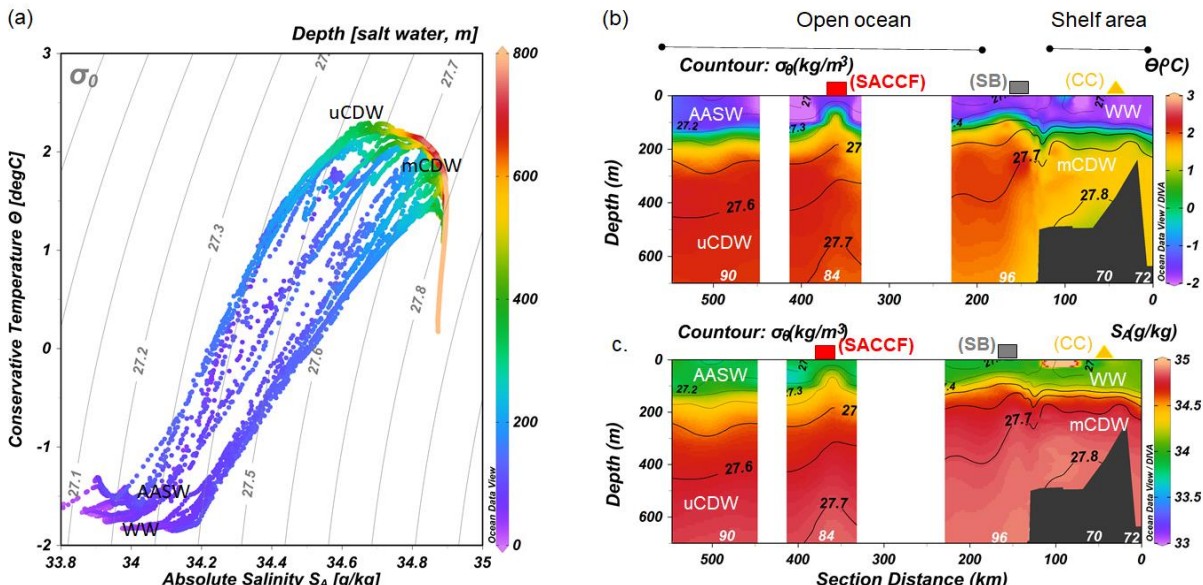

**Figure 2: (a) Diagram of absolute salinity (SA) versus conservative temperature (Θ) with isopycnal lines and colors denoting depth in m. The distribution along the transect shown in Figure 1 of (b) Θ and (c) SA with density (σθ) as contours. The values of Θ and SA were generated by ODV software from CTD data.**

During the austral spring sampling period, Winter Water (WW; $\Theta$ < -1.8 °C and $S_A$ ~34.1) still existed in the upper 100 m at stations in the shelf sea (Figures 2b and 2c). In spring, Antarctic Surface Water (AASW) forms that has a higher temperature ( -1.5 – 1°C) and a lower salinity (33.0 > $S_A$< 33.7) than WW (Tomczak and Godfrey, 2003; Orsi et al., 1995). During sampling, AASW was present at stations outside the shelf region (St. 84, 90 and 96; Figures 2b and 2c). The position of the SB marks the southern terminus of water with CDW properties. From the shelf break toward the open ocean, upper Circumpolar Deep Water (uCDW) existed at 300 m depth, characterized by $\Theta_{max}$ = 2°C and a maximum of $S_A$ ~34.7. Near the WAP, the Antarctic Slope Front is missing (Klinck et al., 2004; Moffat and Meredith, 2018); hence, there is no barrier in the outer shelf region (Klinck et al., 2004). As a consequence, the shelf region is directly affected by the presence of the SB, resulting in subsurface intrusion of uCDW onto the continental shelf. This water mass is modified into cooler and less saline water, referred to as modified CDW (mCDW) (Hofmann and Klinck, 1998), and was present at stations 70 and 72 below 200 m. Ocean eddies, modulated by wind forcing and interaction with the slope, are responsible for the transport of uCDW from the ACC into the inner shelf region. Intrusion of uCDW displaces shelf water, allowing a heat flux to the shelf area that triggers the melting of floating ice shelves. Melting of the glacial ice produces buoyant northward





(offshore) flowing surface water that maintains the continuation of offshore-onshore water mass exchange (Klinck et al., 2004; Moffat and Meredith, 2018), although it did not seem to occur along our transect.

The ice-coverage diminished with increasing distance from the shore, with stations near-shore (St. 70 and 72) having sea ice concentrations of 100%, and falling to 60% at the shelf break (St. 96). The ice-cover dropped to

20% at St. 84 offshore, whereas St. 90 was ice free.

### 3.2 Fe speciation

Stations located in the continental shelf (St. 70 and 72) generally had higher $[L_t]$ and DFe than stations sampled offshore (St. 84, 90 and 96; Figures 3a and 3b). At the shelf stations, $[L_t]$ varied from 1.23 to 6.43 nM eq. Fe and high $[L_t]$ (>2 nM) was present below 200 m in mCDW (Figure 3a). In particular, at St. 72 nearest to the shore,

$[L_t]$ was higher than 2.5 nM eq. Fe at the surface and reached up to 6.43 nM eq. Fe at 400 m close to the sediment. Concentrations of DFe were <0.6 nM (0.29 - 0.52 nM) in the upper 100 m, but >1 nM (1.11 to 2.26 nM) at depths below 200 m near the sediment (Figure 3b).

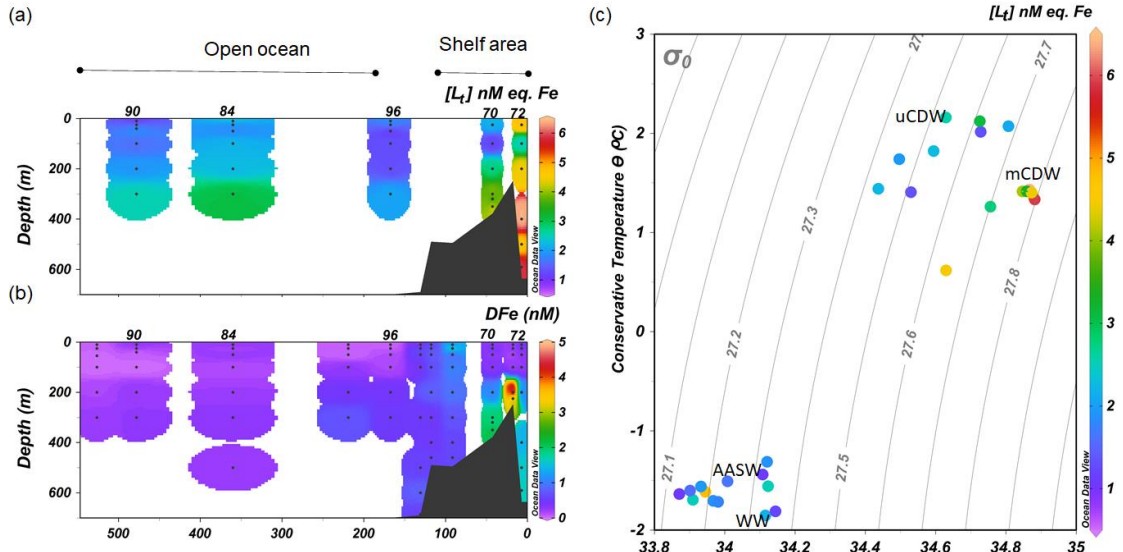

**Figure 3: The distribution along the transect shown in Figure 1 of (a) the concentrations of total Fe-binding ligand $[L_t]$ and (b) concentrations of dissolved-Fe (DFe, data from ; and (c) a Ө- SA diagram with colors denoting depth the values**
**of $[L_t]$.**

Further offshore towards the open ocean (St. 84, 90 and 96), $[L_t]$ varied from 1.07 to 3.09 nM eq. Fe (Figure 3a), whereas DFe concentrations ranged from <0.05 and 0.47 nM (Figure 3b). At the shelf break, approximately where





the SB was located, [$L_t$] was relatively low (St. 96; Figure 3a). Low DFe was observed in the upper 200 m offshore, reaching the lowest concentration (<0.05 nM) at St. 90 farthest from the shelf (Figure 3b).

Two ligand groups were distinguished only in two samples in open ocean waters, both collected at St. 90 (at 40 and 300 m). The measured conditional stability constant for the stronger $L_1$ ligands and the progressively weaker $L_2$ ligands were distinct, and the values did not overlap. At 40 m, log $K_1$ = 12.4±1.1 and log $K_2$ = 10±1.1, whereas at 300 m, log $K_1$ = 11.3±0.6 and log $K_2$ =10.4±0.7. However, the uncertainty for the ligand concentrations was relatively large for the two-ligand model, $L_1$ (0.66±1.81 and 1.58±4.29 nM eq. Fe, at 40 and 300 m, respectively)

and $L_2$ (1.10±4.32 and 1.34±3.74 nM eq. Fe, at 40 and 300 m, respectively), which implies that the one-ligand model fit the data better. Therefore, results of data fitting with the two-ligand model will not be presented.

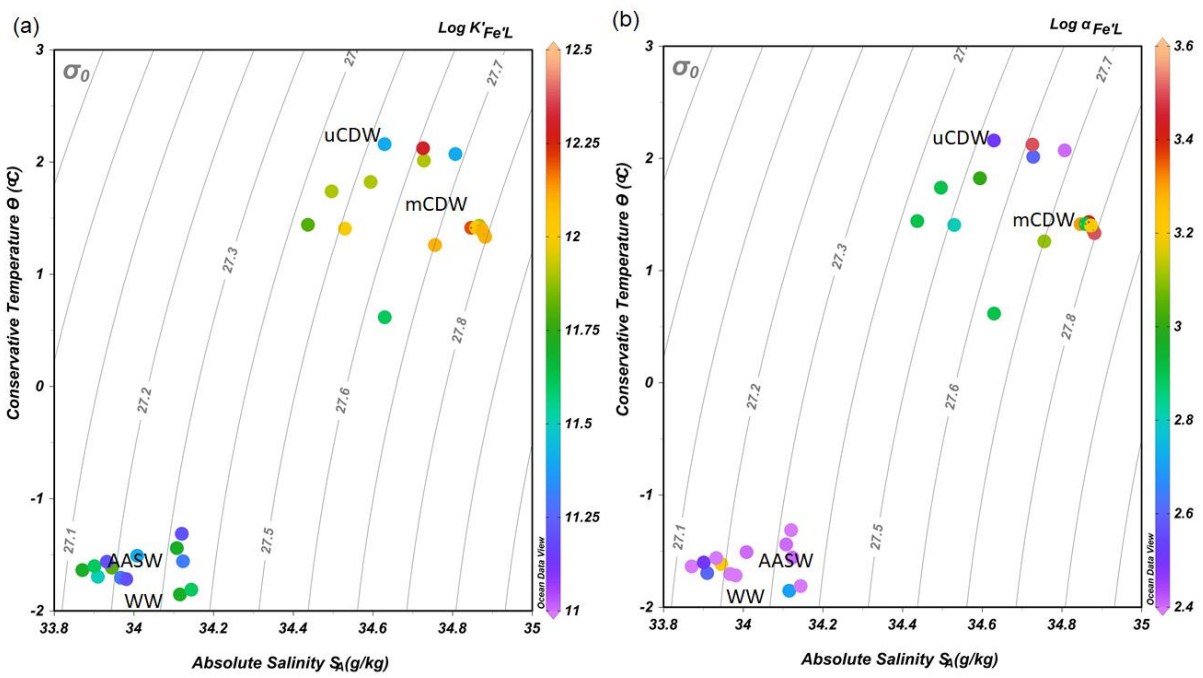

**Figure 4: (a) The binding strength, log $K_{Fe'L}^{cond}$ and (b) complexation capacity, log $\alpha_{Fe'L}$ plotted in a $\Theta$-SA diagram. The color scale indicates the values of log $K_{Fe'L}^{cond}$ and log $\alpha_{Fe'L}$.**

The values of log $K_{Fe'L}^{cond}$ varied within one order of magnitude between 11.1 and 12.3 (Figure 4a). The lower log

$K_{Fe'L}^{cond}$ in AASW coincided with the lowest log $\alpha_{Fe'L}$ (AASW: mean log $\alpha_{Fe'L}$ = 2.6±0.3, N=13; Figure 4b). The largest log $K_{Fe'L}^{cond}$ and log $\alpha_{Fe'L}$ was measured in shelf waters, particularly in mCDW (mean log $\alpha_{Fe'L}$ =3.4±0.2, N=8; Figure 4b).



Ligands were always present in excess of DFe, with [L´], ranging from 0.75 to 4.98 nM eq. Fe. High excess ligand concentrations (>2 nM eq. Fe) were observed close to shore (St. 72) declining towards the shelf break and reaching

the lowest [L´] at St. 96 near the shelf break (Figure 5a). Further offshore (St. 84 and 90), [L´] remains fairly constant at 1 – 2 nM eq. of Fe.

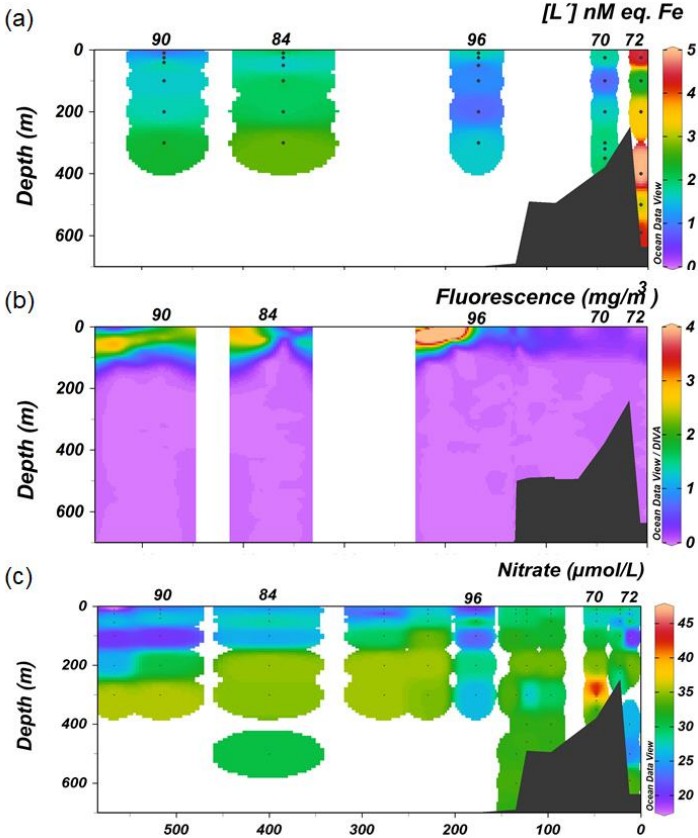

**Figure 5: The distribution along the transect shown in Figure 1 of (a) excess ligand concentrations [L´], (b) Fluorescence, and (c) Nitrate.**



## 4. Discussion


The ligand concentrations measured during our study (1.07 – 6.43 nM eq. Fe; Figure 3a) are consistent with the broad range of Fe-binding ligand concentrations measured in DFe speciation studies in the Southern Ocean (Boye et al., 2001; Lin and Twining, 2012; Nolting et al., 1998; Thuróczy et al., 2011). Previously reported [$L_t$] in the Southern Ocean varies from 0.5 – 1.84 nM eq. Fe in the Atlantic sector (Thuróczy et al., 2011), 2.2 to 12.3 nM

eq. Fe in the Pacific sector (Nolting et al., 1998), and 0.44 – 1.61 nM eq. Fe in the Indian sector (Gerringa et al., 2008). The [$L_t$] in Antarctic polynyas ranges between 0.3 – 1.6 nM eq. Fe (Gerringa et al., 2019; Thuróczy et al., 2012), whereas in regions with sea ice-coverage, [$L_t$] in underlying water is relatively high, with values of 4.9 – 9.6 nM eq. Fe and up to 72.1 nM eq. Fe within the sea ice (Lannuzel et al., 2015; Genovese et al., 2018).

### 4.1 Fe-binding ligands along the transect from the shelf to the open ocean

The high sea ice cover on the continental shelf obstructs the light penetration into the water column, inhibiting the development of an early spring bloom. Therefore bloom generated ligands are less likely to be found. However, microbial excretion from sea-ice algae and bacteria within and just beneath the sea-ice release EPS, which can form Fe-binding ligands (Lannuzel et al., 2015; Norman et al., 2015; Genovese et al., 2018; Hassler et al., 2017). The planktonic community in spring is dominated by diatoms and haptophytes (*Phaeocystis antarctica*) (Joy-

Warren et al., 2019; Arrigo et al., 2017). According to Lannuzel et al. (2015), the omnipresence of tube dwelling diatoms (*Berkelaya* sp.) attached via EPS to the bottom of the sea-ice was responsible for relatively high [$L_t$] in under-ice seawater, indicating that EPS could elevate seawater [$L_t$] in areas of sea-ice cover. In addition, a laboratory study has shown that cultured *P. antarctica* appears to excrete EPS in relatively high concentrations (Norman et al., 2015), with similar binding strength (log $K^{cond}_{Fe'L}$ 11.5 – 12) to those measured in this study (log

$K^{cond}_{Fe'L}$ 11.1 – 12.3). At the ice-covered shelf stations, the high [$L_t$] in the upper water column implies that ice algae exudates are a source of Fe-binding ligands (St. 70 and 72; Figure 3a). Further from the coast, EPS from phytoplankton likely provide an additional surface source of ligands.

High [$L_t$] (>2.75 nM eq. Fe) was observed in mCDW (Figure 3c) with a narrow range of log $K^{cond}_{Fe'L}$ (11.1 - 11.9; Figure 4a), which suggests that similar chemical characteristics and a common origin. Given that uCDW at the

shelf break has [$L_t$] lower than 2.5 nM eq. Fe and relatively low DFe concentrations, the high [$L_t$] in mCDW appears to be supplied by the shelf sediments. Upwelling and contact with the sediment of uCDW presumably results in the resuspension of organic matter and pore water from the sediment, supplying ligands and DFe to the



water column. Indeed, ligand input in the proximity of sediments was previously observed in upwelling regions over the continental shelf or in coastal areas (Gerringa et al., 2008; Buck et al., 2017). Subsequent upwelling

processes may transport the ligands to the upper water column, including rDOM. Moreover, intrusion of uCDW also provides heat (Smith et al., 1999), which may cause glacial and sea ice melt. The melting of sea ice (i.e. first year pack ice) supplies ligands to surrounding seawater (Genovese et al., 2018), whereas glacial ice is not expected to contribute organic ligands. The estimated flux of [$L_t$] from melting pack ice in spring can reach up micromolar levels per square meter of pack-ice per day (Genovese et al., 2018), showing the significance of the sea-ice melt

contribution to the ligand pool. Overall, the high [$L_t$] in the shelf region (Figure 3a) can be explained by several ligand sources associated with sea-ice, including the melting of sea-ice, as well as sediment resuspension and upwelling.

At the continental shelf break (St. 96) in the vicinity of the SB, the lowest [$L_t$] was found in the upper 200 m of the water column. The presence of the SB is noticeable by the increased upward tilt of the isopycnals (Figure 2a)

(Orsi et al., 1995; Klinck et al., 2004). Here the ACC interacts with the continental slope (Orsi et al., 1995), propagating ocean eddies that subsequently cause cross-shelf water intrusion (Moffat and Meredith, 2018). The subsurface intrusion of uCDW and its associated turbulence may cause vertical water mass mixing at the proximate location of the SB. The little ice-cover at the shelf break compared to the inner shelf allows more light penetration, triggering a bloom, as indicated by fluorescence maxima observed at St. 96 (Figure 5b). The bloom

and its related microbial activities could release Fe-binding ligands. However, given the consistently low and constant distribution of [$L_t$] at the shelf break, it seems that mixing determines the distribution and net concentrations of ligands (Figure 3a). This is confirmed by the relatively constant distribution of DFe and macronutrients (i.e. nitrate; Figure 5c) at the same station, indicating that prominent mixing at the shelf break indeed is the major factor governing the distribution of ligands, DFe and nutrients.

Further oceanward from the shelf break, [$L_t$] was >1 nM eq. of Fe (St. 84 and 90; Figure 3a), probably related to the spring bloom at this location. Satellite-based data (Arrigo et al., 2017), showed that open water formed one month earlier offshore than near-shore (St. 70 and 72), implying that the melting of sea ice offshore (St. 84 and 90) occurred preceding and during our occupation. The melting of sea ice released nutrients and micronutrients such as Fe (Lannuzel et al., 2016; Sherrell et al., 2018), which together with the availability of light stimulated

the spring bloom. Such a bloom in turn is a source of Fe-binding ligands in the upper water column (Gledhill and Buck, 2012; Boye et al., 2001; Croot et al., 2004; Gerringa et al., 2019). Arrigo et al. (2017) reported that a bloom in its early stages was observed underneath variable sea ice cover seaward from the shelf break. Indeed, the fluorescence maximum observed in the upper 100 m at the offshore stations (St. 84 and 90; (Figure 5b) concurred



with depletion of DFe and drawdown of macronutrients (N and Si), illustrating the presence of a bloom.

Siderophores are expected to be produced upon Fe depletion by marine microbes as a strategy to acquire Fe (Butler, 2005; Buck et al., 2010; Mawji et al., 2008; Velasquez et al., 2011). Similarly, as detailed above for the shelf stations, the exudation of EPS from diatoms and haptophytes could be an important addition to the organic ligand pool. Moreover, polysaccharide ligands will be released by microbial cells during the bloom as well as via grazing (Sato et al., 2007; Laglera et al., 2019b) and viral lysis (Poorvin et al., 2011; Slagter et al., 2016).

Additionally, the ratios of labile particulate Fe to labile particulate Mn (Seyitmuhammedov, 2020) indicate that Fe has a biogenic origin in the offshore waters. Therefore, we suggest that the origin of $[L_t]$ offshore was, next to the melting of sea ice, the result of *in situ* production of organic ligands during the bloom and passive generation from microbial processes associated with the bloom.

The lowest concentrations of DFe (<0.05 nM) were observed at St. 90 and 84 and were a result of both biological

uptake and limited supply. This area appears to represent Fe-limited conditions as indicated by declining Si* (Si*= [Si] – [N]) values and high ratios of [nitrate]/DFe (Figures 6a and 6b).

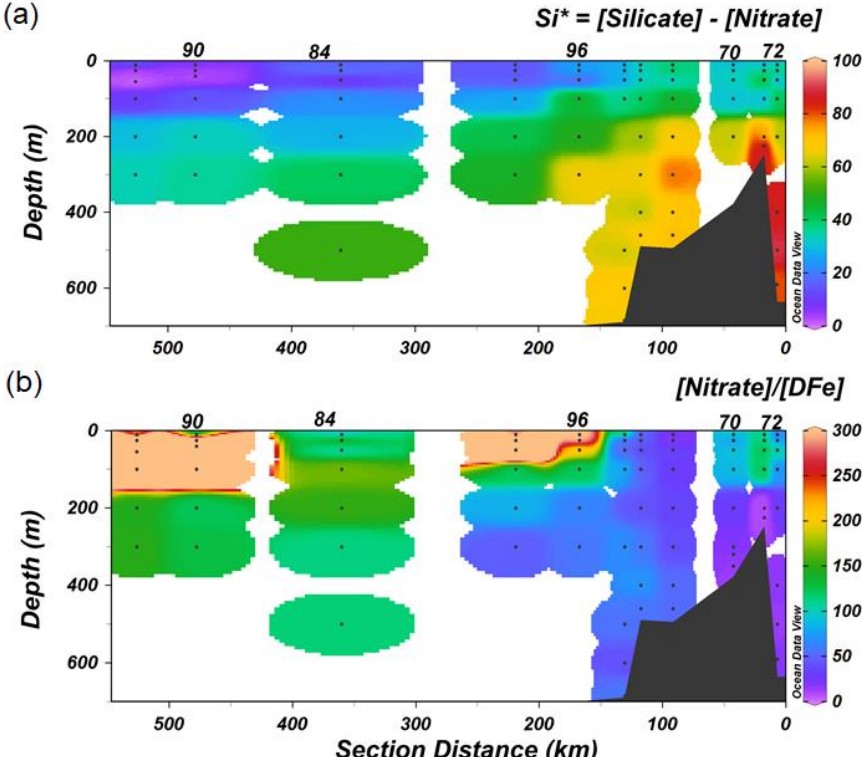

**Figure 6: The distribution of Si*(a) and the ratio of [Nitrate]/DFe (b) along the transect shown in Figure 1.**





The value of Si* serves as a proxy for Fe limitation, where Fe stress leads to preferential drawdown of Si compared to N by diatoms in surface water (Takeda, 1998). A negative Si* indicates Fe limiting conditions, assuming that

Si and N are required in a 1:1 ratio by diatoms (Brzezinski et al., 2002). Typically, organic ligands excreted under Fe-limited conditions have strong affinity for Fe (Maldonado et al., 2005; Mawji et al., 2008), i.e. a high log $K_{Fe'L}^{cond}$ (>12). However, a relatively low log $K_{Fe'L}^{cond}$ is observed in AASW relative to deeper uCDW and mCDW (Figure 4a). This indicates that in offshore AASW where Fe limitation is expected, the contribution of siderophores is modest. Indeed, recent studies showed that only <10% of Fe is complexed by siderophores (Boiteau et al., 2019;

Bundy et al., 2018), suggesting that the binding strength of the overall ligand pool is not always a good indicator of the presence of particular ligand group if multiple ligand sources are present. Moreover, in the presence of light, organic ligands can undergo photo-degradation(Hassler et al., 2020), and thus the chemical structure can be altered into a slightly weaker ligand type (Barbeau et al., 2001; Powell and Wilson-Finelli, 2003). Mopper et al. (2015) suggested that the absorption of solar radiation by chromophoric dissolved organic matter as part of the

ligand pool which commonly produced by sea ice algae   (Norman et al., 2011), leads to the photochemical transformation of these compounds. These photo-oxidative processes can thus also explain the shift in log $K_{Fe'L}^{cond}$ in the AASW to lower values compared to deeper uCDW and mCDW.

The presence of the SACCF and SB fronts affects bloom conditions. As shown by Arrigo et al. (2017), high chlorophyll-a concentrations were observed in surface waters in between the SB and SACCF, which suggest the

distribution of phytoplankton biomass is affected by physical processes in the area. The SB also appears to mark the boundary between offshore organic ligands that result from a combination of the earlier sea ice melt and *in situ* production and/or generation associated with offshore blooms, and organic ligands on the shelf that result from a combination of ice-algae exudation, sea-ice melt, and sediment resuspension. In the region near the SB at the shelf break, water mass mixing due to the baroclinically unstable water column seems to have caused

consistent distributions of [$L_t$] (Figure 3a). Further offshore, ligands are most likely associated with the bloom, but the distribution of ligands is also affected by enhanced vertical mixing and intensified currents proximal to the area of the fronts. Solar radiation enhances stratification and drives the formation of the spring bloom whereas deep mixing can both hinder as well as stimulate bloom formation based on the balance between availability of light and nutrients (Arrigo et al., 2017). It thus seems likely that the balance between mixing and stratification

results in variable [$L_t$] in the area around the SACCF (St. 84 and 90; Figure 3a).





## 4.2 Implications for primary productivity

In general, we found $[L']$ >0.75 nM eq. Fe (Figure 5a), which accounts for approximately 80% of $[L_t]$. This implies that at least 80% of total ligands measured are available to bind Fe, although the total complexation capacity of ligands is also determined by its log $K_{Fe'L}^{cond}$. The highest complexation capacity log $\alpha_{Fe'L}$ was found in mCDW on

the shelf (Figure 4b), and concurred with the highest concentrations of DFe in mCDW (Figure 3b and Figures 4b). The high complexation capacity of ligands on the shelf increases the potential of organic ligands to stabilize additional Fe input to the shelf waters (Lannuzel et al., 2015; Gerringa et al., 2019; Thuróczy et al., 2012) and lengthen the residence time of DFe (Gerringa et al., 2015). A longer residence time has a positive feedback on the development of local primary productivity upon sea ice melting (Arrigo et al., 2017), supplying DFe to

phytoplankton on the shelf. Moreover, based on the results of oxygen isotope ($^{18}$O/$^{16}$O, conventionally reported into delta-notation as $\delta^{18}$O) analysis (Seyitmuhammedov, 2020), meltwater associated with runoff and glacial discharge is present in the upper 200 m of the shelf, and probably is a source of particulate and dissolved Fe that will increase under continued climate change. However, whether Fe in particulate form will partition into the dissolved pool via ligand driven dissolution of Fe, also depends on the fraction of labile particulate Fe. In addition,

local primary productivity not only relies on the DFe input from, for example, meltwater and glacial debris (Klunder et al., 2011; Lannuzel et al., 2016), but probably also on the input of Co and Mn (Saito et al., 2010; Wu et al., 2019; Middag et al., 2013) as these elements showed co-limitation in the Southern Ocean (Middag et al., 2013; Saito and Goepfert, 2008).

Besides affecting the shelf conditions, ice melt also produces buoyant northward-flowing surface water, which

may facilitate DFe transport from the shelf to the open ocean, supplying DFe for primary production offshore, but this effect was not noticeable in the transport data for this specific transect. However, the conditions along the WAP are not homogenous and elevated Fe (dissolved and total-dissolvable; (Seyitmuhammedov, 2020) concentrations northeast of our transect were observed in the upper 100 m, suggesting that some of the observed ligands might have been transported southwesterly with the CC. This high DFe stabilized by organic ligands will

probably be transported further to the southwest where a coastal polynya is commonly observed in Marguerite Bay (Arrigo et al., 2015). Such transport would supply DFe to the highly productive Marguerite Bay polynya and fuel a phytoplankton blooms in these ice-free waters but could also be partly transported offshore in the region southwest of our transect. However, the relative amount of DFe bound to organic ligands can vary, and is also strongly influenced by the continued change in environmental conditions due to global warming (Slagter et al.,

2017; Ye et al., 2020), making it likely such a transport of DFe to the southwest or offshore will change as well.





Global warming has caused glaciers to retreat and induced significant loss of sea-ice, particularly in the Antarctic Peninsula area (Henley et al., 2019; Stammerjohn et al., 2012; Turner et al., 2020). The sea ice extent over the southern Bellingshausen Sea, has decreased in recent decades, creating open water and lengthening the ice-free season (Turner et al., 2013). This results in increased solar irradiance and enhanced stratification (Henley et al., 2019), which can lead to an alteration of the phytoplankton community structure. As previously reported, variable light conditions favor the growth of *Phaeocystis antarctica* over diatoms (Joy-Warren et al., 2019; Alderkamp et al., 2012). In contrast, smaller-cell diatoms are better adapted to increased sea surface temperature (Schofield et al., 2017). Changes in planktonic community composition affect net primary production and overall carbon drawdown, which lead to further alteration of the food web and carbon cycling (Schofield et al., 2017; Joy-Warren et al., 2019; Arrigo et al., 1999; Alderkamp et al., 2012). These and other ongoing changes in the food web will also affect production of dissolved organic carbon (DOC) and thus ligands as they form a fraction of the DOC pool (Gledhill and Buck, 2012; Whitby et al., 2020). Generally, one expects that increased DOC production would lead to more ligands, but the binding strength depends on which molecules are formed (Gledhill and Buck, 2012; Hassler et al., 2017). Additionally, intensified light exposure alters log $K_{Fe'L}^{cond}$ by photo-oxidative processes, possibly reducing the complexation capacity and binding strength for Fe (Barbeau et al., 2001; Powell and Wilson-Finelli, 2003; Mopper et al., 2015) as well as the bioavailability (Hassler et al., 2020). Furthermore, complexation capacity is affected by pH, implying that ongoing ocean acidification also influences the speciation of Fe (Ye et al., 2020). Overall, the continued sea-ice melt and glacial retreat can be expected to increase the supply of Fe (Lannuzel et al., 2016), other micronutrients (Co, Mn, etc.), and Fe-binding ligands (Lin and Twining, 2012), but the consequences for their complexation capacity and overall bio-availability of Fe remain elusive. If DFe becomes progressively more available in the Southern Ocean, phytoplankton growth could increase until another process becomes limiting, such as the availability of another micronutrient or macronutrient. Many uncertainties remain, but the changing environmental conditions of the WAP due to climate change will affect marine biogeochemical cycles and influence productivity beyond the Southern Ocean as the Southern Ocean is an important hub in ocean circulation and its waters eventually supply nutrients to other regions (e.g. Middag et al. (2020).

## 5. Summary

Our results indicate that organic Fe-binding ligands in surface water on the continental shelf of the WAP are associated with ice-algal exudates and addition of ligands from melting sea ice. In the water column close to the

continental slope and shelf sediments, resuspension of sediment followed by upwelling processes appears to be
        another source of ligands. From the continental shelf-break oceanward, sources of Fe-binding ligands are likely
        related to offshore phytoplankton blooms, either actively produced during the bloom, or passively generated by
        microbial processes associated with the bloom. The distribution of ligands is affected by the two major fronts in
        the region, the SACCF and SB. The SB along the shelf break not only marks the boundary between the shelf and
open ocean, but also marks the border between organic ligands associated with the bloom offshore and organic
        ligands on the shelf originating from sea-ice and sediment related sources, such as ice-algae exudation, sea-ice
        melt, and sediment resuspension. Overall, excess ligands comprised up to 80% of the total ligand concentrations,
        implying the potential to solubilize additional Fe input. The ligands in denser mCDW on the shelf have a higher
        complexation capacity for Fe, and are thus capable of increasing the residence time of Fe as DFe and fuel local
primary production later in the season upon ice melt.

## 6.   Author contribution

IA and KS performed the analysis. IA and LG prepared the content of the manuscript. RM, SS and CS design the
research and edited the manuscript. KRA and G-JR performed the research and edited the manuscript. All authors
are contributed to the final version of the manuscript.

## 7.   Acknowledgements

The authors would like to thank the captain and his crew of the R/VIB N. B. Palmer, as well as Anne-Carlijn
Alderkamp and all other participants, for their efforts and support. Our colleagues in the nutrients lab at NIOZ are
acknowledged for analyzing nutrients. IA was financed by Indonesia Endowment Fund for Education (LPDP),
and KS received a scholarship from the University of Otago. KRA was funded by a grant from the National
Science Foundation Office of Polar Programs (ANT-1063592). The IAEA is grateful to the Government of the
Principality of Monaco for the support provided to its Environment Laboratories.

## 8.   Data availability

Data will be available at  https://doi.org/10.25850/nioz/7b.b.5





### 9. Competing interest

The authors declare that they have no conflict of interest.

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
