# Peer review of "Fe-binding organic ligands in coastal and frontal regions of the western Antarctic Peninsula"

_Biogeosciences, 2020_

## Referee Comment (RC1) · Anonymous Referee #1 · 6 Dec 2020

General comment: This study investigated Fe-binding organic ligands distribution upper 600-m depth at 5 stations in the western Antarctic Peninsula (WAP). The research area covered the front and southern boundary of the Antarctic Circumpolar Current (ACC) as well as the zone influenced by the Coastal Current (CC) near the peninsula. The results indicated that the organic ligands on the shelf were associated with ice-algal exudates and melting sea-ice in surface water, and those in the deep shelf water were supplied via resuspension of shelf or sediments. The ligands concentration always exceeded dissolved Fe concentration, suggested that any additional Fe input can be stabilized in the dissolved form via organic complexation. Overall, this manuscript is well written and organized. But there are two points to be considered.

(1) The relationship between complexation capacity of the ligands and Fe distributions

The authors explained the relationship between complexation capacity of the ligands and Fe distributions in the beginning of section 4.2, but it was about the specific sample. How was the overall trend?

(2) The meaning of excess ligands In this study area, [L'] was always observed and additional Fe input was expected to be stabilized in the dissolved form. Although particulate Fe was not investigated in this study, it was expected that some portion of Fe might exist as particulate form in the WAP (Seyitmuhammedov, 2020). The co-existence of [L'] and particulate Fe sounds like a contradiction. How do the authors think about the contradiction? But I could not access the reference Seyitmuhammedov (2020) via online because it is Doctoral thesis; so I'm not sure whether Seyitmuhammedov (2020) researched the total dissolvable Fe during the same cruise to this study. If so, please explain brief results from Seyitmuhammedov (2020).

It is well recognized that the Fe speciation data in the ocean is important to understand Fe cycle in marine environment, the result and finding obtained in this study are valuable for future studies. Several minor comments are listed below.

Minor comments: Page 2, L38. CO2 "2" should be written in subscript.

Page 3, L72-79. Humic substances (HS) and HS-like substances. . . Complicated notation. Because this study did not investigate the HS and HS-like substances specifically, the authors can unify the terms and explain in this section.

Page 4, L114-115. Low density polyethylene bottle (LDPE, Nalgene). In general, GEO-TRACES cookbook recommends fluorinated high density polyethylene bottle (FLPE) or Teflon bottles for the sampling of ligands in order to avoid the absorption to the bottle wall. Did the authors check the influence of the difference on the CLE-AdCSV?

Page 6, L137-. Section 2.3 Did the authors apply air purge method? Please add the information about the purging method.

Page 6, L158-159. . . .the product of [L'] and log K,. . . Probably the authors can elimi-

nate "log" from the sentence.

Page 7, L164-165. The conditional stability constant of.... Did it mean that different calibration result from the original method (Abualhaija and van den Berg, 2014) was obtained?

Page 8, Figure 2 (b)and(c). Please add the boundary line between mCDW and uCDW in Figures 2 (b) and (c).

Page 9, Figure 3 (a), (b) and (c) Please add the titles for x-axis.

Page 11, Figure 5 (b) and (c). Please add the data points in the Figures 5 (b) and (c), too.

Page 12, L249- Section 4.1 Why there was the huge differences in [L'] distributions in deeper water between stations 70 and 72? Both stations are located in the shelf region but separated by a sill. It is very interesting. In the deeper waters at station 72, high Si* and low N/DFe values were observed. Is [L'] likely to have a relationship with Si* or N/DFe?

Page 13, L285 However, given.... I think the mixing process influenced on the distribution of phytoplankton as well as on those of Fe, L and nutrients. I think the ligand production rate by phytoplankton is different between species and their physiological status, too.

Page 16, L341-. Section 4.2. Are there any information about the phytoplankton species during this observation?

---

## Referee Comment (RC2) · Anonymous Referee #2 · 24 Dec 2020

The study reports an investigation on the organic speciation of Fe along a natural gradient of the western Antarctic Peninsula. Although there were few stations investigated, the overall data can provide valuable information on the role of Fe and its ligands in a crucial area of the planet. The manuscript is generally well written and clear to follow.

I have some major remarks (mainly related to the presence and the quality of the data), along with some minor comments.

Major comments

1) I do not completely agree with the use of the reference Seyitmuhammedov 2020, being a PhD thesis not available. If it had been used just for a minor aspect, it would have been ok, but it is often cited, particularly for data that are present there but not

presented in this manuscript. First of all, I think some additional detail for the DFe analysis (section 2.2) could be useful and I suggest to add them. However, the main problem is related to the values of labile particulate Fe and Mn (section 4.1), $\delta18O$ and dissolved and total-dissolvable Fe (section 4.2). In order to help readers, I think that they could be presented at least with ranges. Maybe it could have been smoother to publish those values before submitting this manuscript, to have a proper reference to cite.

2) I looked at the dataset presented in the reported link (https://doi.org/10.25850/nioz/7b.b.5) and I have some questions or remarks with the presented data and their use in the Results or Discussion sections.

2a. Fluorescence. What do negative values for fluorescence mean? Are they just a consequence of improper calibration or do they have another meaning? In addition, there are some fluorescence data missing (two depths for Station 70 and all the depths for Station 72), hence I wonder how the plots were drawn for Figure 5b. Please clarify.

2b. DFe. Are data for St 90 40 e 100 m below the LOD? I ask that because that there is no standard deviation for those parameters, and also because the standard deviation of the blanks is reported as 0.02 nM (line 134), hence the LOD should be around 0.06 nM by using the $3\sigma$ method, which is higher than the values reported for those two samples (0.05 nM). If so, I think it should be clearly expressed, but in that case I wonder how the values could be plotted in Figure 3b (maybe as half the LOD?) and how the CLE-AdSV analyses were performed for those two samples, since they would need a value of DFe for the voltammetric titration. Please clarify this aspect and make the corrections if needed.

2c. Silicate. Why data for Silicate are not reported in the table? Also, in line 314, to express the purpose of the Si* values, the authors comment that "a negative Si* indicates Fe limiting conditions", but in their dataset there are no negative values for Si*. Please explain better this point.

Minor comments

- Line 38: correct CO2 ("2" in subscript).

- Section 2.1: please define the material of the 0.2 $\mu$m filters used for filtration and the volume of the GO-FLO bottle. Although the conservation procedures are correct, I wonder why the samples for Fe-binding ligands and DFe were collected separately, instead of freezing just one bottle and take the aliquots for the two analyses from the same "container" in the lab (of course acidifying before DFe analysis).

- Figure 1: I suggest using a darker yellow to indicate the Coastal Current.

- Section 2.2: please report the certified or informative values of SAFe D1 and GSP samples. In addition, report also the LOD of the procedure.

- Line 153: in "CLE-CSV" there is an "Ad" missing before "CSV".

- Line 156: the full stop at the end of the sentence is missing.

- Line 158: please close the parenthesis which was opened before "$\alpha Fe\hat{I}\check{D}L$".

- Line 160: the authors refer to $\alpha FeL$, but I guess they meant $\alpha Fe\hat{I}\check{D}L$ instead?

- Figure 2: please uniform the indication of "c." for the third figure, using the two parentheses consistently with (a) and (b). Also, in the caption, the "$\theta$" in "$\sigma\theta$" should be in subscript. Finally, Absolute Salinity is reported with "A" in subscript or as plain SA in the text and in the Figures, please uniform in the whole manuscript.

- Line 189: I think there is some problem with the "<" and ">" for Absolute Salinity. Did the authors mean "33.0 < SA < 33.7"?

- Figure 3: there is a reference missing (and an unclosed parenthesis) in "DFe, data from ;". Also, what do the author mean when they say "with colors denoting depth the values of [Lt]"? I guess there's a "depth" in excess?

- Line 233: please remove the comma after [L'].

[Figure]

- Figure 5: why in some images the profiles are "smooth" (e.g. b) and in others are "rounded" (e.g. a and c)? Also, in Figure 5a there are only the profiles for the 5 stations, well separated, while for example in Figure 5c there are more. Why?

- Line 264: I think the "that" is in excess?

- Line 284: since it is one value, it should be "maximum", while "maxima" is used for plurals (accordingly, correct also line 298 from "maximum" to "maxima" if it is referred to more than one).

- Line 298: unclosed parenthesis in "(St. 84 and 90; (Figure 5b)".

- Figure 6: please insert the unit of measurement for Si*. Moreover, in the Figure there is "[Nitrate]/[DFe]" while in the caption there is "[Nitrate]/DFe", please uniform (DFe is presented without parentheses in the whole manuscript).

- Line 325: please revise the "which commonly produced by" part, I do not think the sentence is fluid.

- Line 367: "a phytoplankton blooms": it should be either "a phytoplankton bloom" or "phytoplankton blooms", please correct.

- References are not well uniform in the use of the doi. Maybe there are also some parts missing (e.g. pages or article number for Arrigo 2008, Lam 2011, Lannuzel 2016, etc.).

---

## Referee Comment (RC3) · Anh Pham (Referee) · 27 Dec 2020

In this manuscript, Ardiningsih et al. report measurements of the dissolved iron (Fe) and organic ligands concentrations, as well as the ligand conditional binding strength, in the upper 600-m water along five stations in the western Antarctic Peninsula (WAP) region of the Southern Ocean, where iron limits the ocean primary production. This region shows distinct features in the dissolved Fe pattern and in the hydrographic dynamics. It also contains biological hotspots in regions close to the shelf sea, where the marine ecosystem can be directly impacted by climate change. Thus, it is important to understand the mechanisms controlling the distribution of organic ligands and dissolved Fe in the WAP. The results of this manuscript suggest that ligands in the surface water of the shelf regions are the products of ice-algal exudates and sea-ice

melting, while ligands in the deeper shelf water are produced from sediment-related processes. In the open ocean water of the WAP region, ligand productions are likely to be related to the sea-ice melting process and to the phytoplankton bloom. Mover, the authors show that the ligands concentration in the WAP always exceeds dissolved Fe concentration. This result suggests that any additional Fe input can be stabilized in the dissolved form, thereby being bioavailable for phytoplankton. Overall, this manuscript is well written and easy to follow. However, I would like to raise some questions and suggestions regarding the title, introduction, and discussion of the manuscript, which hopefully can be considered by the authors. I would be more than happy to backdown from these questions if the authors do not agree and provide good counter-arguments.

Major questions/comments

(1) First, the title (Sources of Fe-binding organic ligands in surface waters of the western Antarctic Peninsula) and the introduction of the manuscript give me the impression that this study will use a new method/technique other than CLE-AdCSV to identify the sources of organic ligands in the WAP region. In the Introduction, the authors wrote "The application of AdCSV gives the total concentration ($[Lt]$) and conditional binding strength of the dissolved organic ligands but does not provide information on the identity of ligands" (line 56-57) then stated that "the sources and identities of Fe-binding ligands are still largely unknown" (line 58=59). Thus, I was excited to see what (new) methods the authors would use to pinpoint the sources of organic ligands in the WAP region, which is a very important issue to address, and I think, have not been done before. However, at the end of the Introduction, the authors wrote (line 97-100): "In order to probe sources and distributions of Fe-binding ligands along a natural gradient of Fe, the CLE-AdCSV technique was used to quantify the total concentrations and conditional stability constants of Fe-binding ligands". To be honest, I was a bit disappointed and confused at this point. As an ocean biogeochemistry modeler who does not have a strong background on measurement techniques, I do not understand how the authors can probe the sources of ligands by using the CLE-AdCSV technique, which was

stated before that cannot be used to provide information on the identity of ligands. It turns out that, if I understood correctly, the authors used CLE-AdCSV to measure the concentration and strength of organic ligands, then they based on other hydrographic and biogeochemical features, as well as previous studies, to hypothesize/argue about the potential origins of the measured ligands. While their arguments are valid, I think it is different from measurements that directly pinpoint to ligands' origin and identity. Thus, I would suggest the authors to modify the title and introduction such that they are not misleading and reflect correctly the problems that the manuscript directly address and the methods that the authors use to achieve this goal. Again, in my mind, this is a study that measures the ligands concentration and binding strength, then suggest their potential sources, not a study that directly identify the sources of ligands.

(2) Second, in section 4.2 of the manuscript, the authors discussed at length on how a high complexation capacity of ligand and ice-melting processes can control the ocean primary productivity in the WAP region. They also discussed on the potential impact of global warming on Fe chemistry and ligand, and stated that (line 387-390): "Overall, the continued sea-ice melt and glacial retreat can be expected to increase the supply of Fe (Lannuzel et al., 2016), other micronutrients (Co, Mn, etc.), and Fe-binding ligands (Lin and Twining, 2012), but the consequences for their complexation capacity and overall bio-availability of Fe remain elusive." But what about the impact of ligand production from ice-algal exudates, sediments, and phytoplankton bloom? How will these processes change in the future under the impact of global warming? Is the ligand production from ice-algal exudates going to increase or decrease with sea-ice melting? Since these are major processes in producing ligands, I would love to see more discussion on them.

Minor comments:

Line 16: Our results indicate that organic ligands in "the" surface water. . .

Line 17: Organic ligands in "the" deeper shelf water

Line 41: Should the reference here be Henley et al., 2019?

Line 69-70: covered by sea-ice (remove with)

Line 85-86: such as glacial meltwater, sediments", and" upwelling

Line 171: hydrographic features of the WAP "was" described elsewhere

Line 173: Two distinct horizontal currents exist in the study area: The Coastal Current (CC) and the Antarctic Circumpolar (replace, by: )

Figure 3 caption: remove depth in colors denoting depth the values. . .

Line 323-326: Maybe revise this sentence to make it shorter and clearer: "Mopper et al. (2015) suggested that the absorption of solar radiation by chromophoric dissolved organic matter as part of the ligand pool which commonly produced by sea ice algae (Norman et al., 2011), leads to the photochemical transformation of these compounds."

---

## Referee Comment (RC4) · Jonathan Lauderdale (Referee) · 31 Dec 2020

Jonathan Lauderdale (Referee)

jml1@mit.edu

In this manuscript, Ardiningsih and coauthors present biogeochemical observations from a cruise transect from the western Antarctic Peninsula offshore into the Southern Ocean. This region is particularly susceptible to changes in climate. Three different hydrographic regions are identified, influenced by watermass type and sea ice cover that are suggested to host different distributions and characteristics of dissolved iron and organic iron-chelating ligands: firstly, surface "winter waters" near the coast and on the continental shelf are strongly influenced by sea ice cover with organic ligand production associated with ice-associated algae and iron supply from glacier melt; secondly, upwelling deep waters on the continental slope are initially low in iron and ligands, but concentrations increase as a result of sediment-water interactions and resuspension; and thirdly, in offshore waters of the Antarctic Zone influenced by seasonal sea ice melt, phytoplankton blooms deplete nutrients and iron, while actively or passively producing organic ligands.

I thought this paper was logically organized and engagingly written, and I think the authors do a good job of balancing the fact that the CLE-AdCSV methodology gives information about how much and how strong the ligand(s) present are but not what compounds, with some well-reasoned evidence-based assumptions. I was particularly interested to read about the tube-dwelling sea ice diatoms, and the interaction between sediments and upwelling circumpolar deep waters.

However, I thought the presentation of results was not optimal. I did find the TS diagrams very informative – nevertheless, all the figures had some issues: 1.) The yellow coastal current arrow blends in to the pale blue bathymetry in Figure 1 (also the "CC" triangle in Fig.2). I would have liked to see actual surface current data (maybe LADCP or something?) at the station locations, and how they compare to the broad-brush arrows used to mark the ACC jets. 2.) Figures 2-6 use a rainbow color palette that is firstly not universally accessible to those with color vision deficiency, and secondly creates a perceptually non-uniform color space that can create misleading gradients in continuous data such as temperature and salinity, as well as dissolved iron and ligand concentrations (e.g. https://blogs.egu.eu/divisions/gd/2017/08/23/the-rainbow-colour-map/ or https://www.nature.com/articles/s41467-020-19160-7). 3.) The use of "Ocean Data View" section plots is an issue for me, both because of the colormap, but also because of the variable size of each colored "blob" for bottle data. In Fig.5a for example, station 84 has wide circles while station 90 has narrower circles (and 96 has even narrower), yet it is not true that station 84's concentrations are applicable on a larger spatial scale than station 90's (or even station 96's) – the fluorescence data in Fig.5b clearly has much shorter spatial variability (from more stations/continuous CTD cast data?) that is similar at all stations. ODV's "patchiness" really emphasizes the sparsity of the ligand and iron measurements here, which is a shame to detract from a precious

dataset that clearly has interesting information within it – perhaps horizontally-stacked line profiles might work better than contours? As I said, in general I found the TS plots were more interesting and useful.

Finally, a minor point is that I found the station ordering to be a bit confusing because they are presented in chronological order, while the manuscript suggests an order that is more geographically orientated. I wonder whether the authors could use an alphabetical scheme for this paper with a key in Fig.1 relating the letters back to station number for posterity (so, the stations would be A (72) to E (90) according to distance from the coast)?

---

## Author Comment (AC4) · 21 Jan 2021

We thank the referee for these kind words, the thoughtful comments and efforts towards improving our manuscript.

1) I do not completely agree with the use of the reference Seyitmuhammedov 2020, being a PhD thesis not available. If it had been used just for a minor aspect, it would have been ok, but it is often cited, particularly for data that are present there but not presented in this manuscript. First of all, I think some additional detail for the DFe analysis (section 2.2) could be useful and I suggest to add them. However, the main problem is related to the values of labile particulate Fe and Mn (section 4.1), $\delta$18O and dissolved and total-dissolvable Fe (section 4.2). In order to help readers, I think that

they could be presented at least with ranges. Maybe it could have been smoother to publish those values before submitting this manuscript, to have a proper reference to cite.

Reply: We have added additional detail for DFe analysis in section 2.2 (lines 135 - 149):

"The DFe analysis is described in detail by Seyitmuhammedov et al. (in review). In short, the DFe analysis was conducted using high‐resolution inductively coupled plasma mass spectrometry (HR-ICP-MS) using a Thermo Fisher Element XR instrument at NIOZ, the Netherlands and using an Amtek Nu Attom instrument at the University of Otago, New Zealand. Samples were UV-oxidized and pre-concentrated using an automated seaFAST system (SC‐4 DX seaFAST pico; ESI) equipped with Nobias-PA1 chelate resin. The quantification was done via standard additions. The recovery of the resin was ∼100% and was verified in every analytical run by comparison between the slope of the seawater calibration curve and the eluent acid calibration curve after (Biller et al., 2012). Accuracy and reproducibility were monitored by regular measurements of the reference materials SAFe D1 and GEOTRACES South Pacific (GSP) seawater, and an in-house reference seawater sample, North Atlantic Deep Water (NADW). Results for DFe analyses of reference samples were 0.722 ± 0.008 nM (n = 3; NIOZ) and 0.729 ± 0.018 nM (n = 6; U. Otago) for SAFe D1 2013 (consensus value = 0.69 ± 0.04 nM) and 0.155 ± 0.045 nM (n = 13) for GSP 2019 consensus values. The average overall method blank (seaFAST and ICP-MS), determined by repeatedly measuring acidified ultrapure water in every analytical run as a sample, was 0.05 ± 0.02 nM (n = 21)."

The ratio of labile particulate Fe and Mn in the open ocean was 0.27 ± 0.49, and has been added in the text (line 326):

"Additionally, the ratios of labile particulate Fe to labile particulate Mn (0.27 ± 0.49; Seyitmuhammedov et al. (in review) indicate that Fe has a biogenic origin in the offshore waters (Twining et al., 2004)." the range of the $\delta$18O results from oxygen isotope analysis was added in lines 373 – 375. "the results of oxygen isotope (18O/16O, conventionally reported into delta-notation as $\delta$18O; Seyitmuhammedov et al. (in review) analysis showed $\delta$18O values ranged from -0.56 – 0.06 ‰These values were used to with estimated fractions of sea-ice meltwater (-1.9 – 1.1 %) and meteoric meltwater (precipitation and glacial;0.3 – 3.9 %)."

The range of DFe and total-dissolvable Fe in the study region during the sampling period was added in lines 386 – 387):

"the conditions along the WAP were not homogenous and elevated Fe (ranged from 0.08 – 4.88 nM for DFe and 0.16-85.42 nM for total-dissolvable; (Seyitmuhammedov, in review)) concentrations northeast of our transect were observed in the upper 100 m, suggesting that some of the observed ligands might have been transported southwesterly with the CC." The values of labile particulate Fe and Mn (section 4.1), $\delta$18O and dissolved and total-dissolvable Fe will be available separately upon publication of Seyitmuhammedov et al. (in review). Similarly, the more detail procedure of DFe analysis will be available in Seyitmuhammedov et al. (in review).

2) I looked at the dataset presented in the reported link (https://doi.org/10.25850/nioz/7b.b.5) and I have some questions or remarks with the presented data and their use in the Results or Discussion sections. 2a. Fluorescence. What do negative values for fluorescence mean? Are they just a consequence of improper calibration or do they have another meaning? In addition, there are some fluorescence data missing (two depths for Station 70 and all the depths for Station 72), hence I wonder how the plots were drawn for Figure 5b. Please clarify.

Reply: Fluorescence data is obtained from a sensor attached to CTD rosette. The negative values only occur in the deeper than 100m where normally phytoplankton concentrations are really low. The calibration equation that is used to convert Volts to chl-a concentration in units of 'mg/m3' apparently creates negative values for the con-

centration when there is basically no chl-a. Thus, the negative values are basically 0 mg/m3. In our data, the lowest negative value is > -0.018, which is close to zero. The maximum fluorescence in the CTD file is about 2.43, and 0.018 is only a small percentage of the max. Finally, we have added the missing data and added the explanation above in the caption of figure 5 and 'read me' text of table.

2b. DFe. Are data for St 90 40 e 100 m below the LOD? I ask that because that there is no standard deviation for those parameters, and also because the standard deviation of the blanks is reported as 0.02 nM (line 134), hence the LOD should be around 0.06 nM by using the $3\sigma$ method, which is higher than the values reported for those two samples (0.05 nM). If so, I think it should be clearly expressed, but in that case I wonder how the values could be plotted in Figure 3b (maybe as half the LOD?) and how the CLE-AdSV analyses were performed for those two samples, since they would need a value of DFe for the voltammetric titration. Please clarify this aspect and make the corrections if needed.

Reply: Yes, DFe at St90 at depth 40 and 100 was below the LOD and these samples are presented as the value of the LOD without an error. This is stated at the end of the paragraph 215. The LOD is 0.05 calculated from 3xSD of the blank (blank SD = 0.016 ; LOD is 0.054 ). However, we reported in 2 significant figures hence 0.05 instead of 0.06 nmol/L. We used the DFe concentration of 0.05 nM for these two samples.

2c. Silicate. Why data for Silicate are not reported in the table? Also, in line 314, to express the purpose of the Si* values, the authors comment that "a negative Si* indicates Fe limiting conditions", but in their dataset there are no negative values for Si*. Please explain better this point

Reply: We have added Silicate data to the table. Based on the cited literature, a negative value of Si* indicates Fe limiting conditions. In our dataset, there were no negative values for Si*, however, some values were close to 0 (e.g at the open ocean station). Our data indicates that although there is no Fe-limitation yet during our sampling period, Fe limitation could potentially occur later in the season. We have added
the explanation above in the lines 326 – 330, as copied below:

"The lowest concentrations of DFe (<0.05 nM) were observed at St. D and E and were
a result of both biological uptake and limited supply. This area most likely represent
Fe-limited conditions as indicated by declining Si* (Si*= [Si] – [N]) values and high
ratios of [nitrate]/DFe (Figures 6a and 6b). The value of Si* serves as a proxy for
Fe limitation, where Fe stress leads to preferential drawdown of Si compared to N by
diatoms in surface water (Takeda, 1998). A negative Si* indicates Fe limiting conditions,
assuming that Si and N are required in a 1:1 ratio by diatoms (Brzezinski et al., 2002).
In our dataset, although there were no negative values for Si*, some Si* values at the
open ocean stations were close to 0. Our data indicates that although there is no Fe-
limitation yet during our sampling period, Fe limitation could potentially occur later in
the season."

Minor comments

- Line 38: correct CO2 ("2" in subscript). Reply:done

- Section 2.1: please define the material of the 0.2 $\mu$m filters used for filtration and
the volume of the GO-FLO bottle. Although the conservation procedures are correct,
I wonder why the samples for Fe-binding ligands and DFe were collected separately,
instead of freezing just one bottle and take the aliquots for the two analyses from the
same "container" in the lab (of course acidifying before DFe analysis). Reply: we have
added the information on the material of the filters and added the volume of GO-FLO
bottle (lines 117 - 120).

"Seawater samples for DFe and Fe-binding ligands in this study were obtained using
12 L GO-FLO bottles attached to a Kevlar$^®$ wire. Seawater samples were filtered over
0.2 $\mu$m filters (cellulose acetate, Sartroban 300, Sartorius$^®$) into pre-cleaned sample
bottles inside a trace metal clean van." DFe samples were acidified immediately on-
board to minimize the adsorption to the bottle wall. Previous Based on previous studies

(i.e Jensen et al. (2020)) and based on our experiences, the DFe concentration from the ligand sample bottles are somewhat lower than the DFe concentration from immediately acidified samples due to precipitation in unacidified samples.

- Figure 1: I suggest using a darker yellow to indicate the Coastal Current. Reply:Done.

- Section 2.2: please report the certified or informative values of SAFe D1 and GSP samples. In addition, report also the LOD of the procedure. Reply: we have included the certified values of SAFe D1 and GSP samples (lines 140 - 141). Also, the LOD of the procedure is added (lines 141 - 142).

- Line 153: in "CLE-CSV" there is an "Ad" missing before "CSV". reply:Done.

- Line 156: the full stop at the end of the sentence is missing. Reply:Done.

- Line 158: please close the parenthesis which was opened before "$\alpha$Fe'L". ЁĞ Reply:corrected.

- Line 160: the authors refer to $\alpha$FeL, but I guess they meant $\alpha$Fe'L instead? Rreply:corrected.

- Figure 2: please uniform the indication of "c." for the third figure, using the two parentheses consistently with (a) and (b). Also, in the caption, the "$\theta$" in "$\sigma\theta$" should be in subscript. Finally, Absolute Salinity is reported with "A" in subscript or as plain SA in the text and in the Figures, please uniform in the whole manuscript. Reply: done.

- Line 189: I think there is some problem with the "" for Absolute Salinity. Did the authors mean "33.0 < SA < 33.7"? Reply: corrected.

- Figure 3: there is a reference missing (and an unclosed parenthesis) in "DFe, data from;". Also, what do the author mean when they say "with colors denoting depth the values of [Lt]"? I guess there's a "depth" in excess? Reply: corrected.

- Line 233: please remove the comma after [L']. Rreply: Done.

- Figure 5: why in some images the profiles are "smooth" (e.g. b) and in others are "rounded" (e.g. a and c)? Also, in Figure 5a there are only the profiles for the 5 stations, well separated, while for example in Figure 5c there are more. Why? Reply: Fe-binding ligand samples are only taken from 5 stations, therefore, in Figure 5a, where [L'] is presented, there are only the profiles from the 5 stations. For nutrient analyses (in this case, nitrate), samples were taken in a few more stations than ligand samples. Similarly, for Fluorescence data, this data is obtained from the sensor attached to CTD rosette, and fluorescence is recorded whenever the CTD is deployed to obtain seawater for many different analyses, thus we have higher resolution data for fluorescence.

- Line 264: I think the "that" is in excess? Reply: corrected

- Line 284: since it is one value, it should be "maximum", while "maxima" is used for plurals (accordingly, correct also line 298 from "maximum" to "maxima" if it is referred to more than one). Reply: done.

- Line 298: unclosed parenthesis in "(St. 84 and 90; (Figure 5b)". Reply: Done

- Figure 6: please insert the unit of measurement for Si*. Moreover, in the Figure there is "[Nitrate]/[DFe]" while in the caption there is "[Nitrate]/DFe", please uniform (DFe is presented without parentheses in the whole manuscript). Reply:done

- Line 325: please revise the "which commonly produced by" part, I do not think the sentence is fluid. the sentence is revised into two sentences. Reply: done.

- Line 367: "a phytoplankton blooms": it should be either "a phytoplankton bloom" or "phytoplankton blooms", please correct. Reply: corrected

- References are not well uniform in the use of the doi. Maybe there are also some parts missing (e.g. pages or article number for Arrigo 2008, Lam 2011, Lannuzel 2016, etc.). Reply: we have checked the references and use the DOI uniformly in each reference. We also checked the journal volume and pages for all the references.

[Figure]

[Figure]

![Map of the sampling sites showing the Western Antarctic Peninsula region with labeled currents SACCF, SB, ACC, CC; Bellingshausen Sea, Weddel Sea, Antarctic Peninsula, Bransfield strait, Adelaide Island, Marguerite Bay; sampling sites D (84), E (90), C (96), B (70), A (72). Depth color scale from 5 m to 5500 m. Ocean Data View.]

**Fig. 1.** : Map of the sampling sites along our study transect near the Western Antarctic Peninsula.

[Figure]

**Fig. 2.** a) Diagram of absolute salinity (SA) versus conservative temperature (Ƨ) with isopycnal lines and colors denoting depth in m. The distribution along the transect shown in Figure 1 of (b) Ƨ and (c) SA

[Figure]

**Fig. 3.** The distribution along the transect shown in Figure 1 of (a) the concentrations of total Fe-binding ligand [Lt] and (b) concentrations of dissolved-Fe (DFe);and (c) a Ƨ- SA diagram

[Figure]

**Fig. 4.** (a) The binding strength, log "K" _"Fe'L" ˆ"cond" and (b) complexation capacity, log αFeÎÐL plotted in a Ƨ-SA diagram. The color scale indicates the values of log K and log αFeÎÐL.

[Figure]

**Fig. 5.** The distribution along the transect shown in Figure 1 of (a) excess ligand concentrations [LĨĎ], (b) Fluorescence, and (c) Nitrate.

[Figure]

[Figure]

**Fig. 6.** The distribution of Si*(a) and the ratio of [Nitrate]/DFe (b) along the transect shown in Figure 1.

---

## Author Comment (AC5) · 21 Jan 2021

Figures

[Figure]

**Fig. 1.** Map of the sampling sites along our study transect near the Western Antarctic Peninsula.

[Figure]

**Fig. 2.** (a) Diagram of absolute salinity (SA) versus conservative temperature (Æ§) with isopy-cnal lines and colors denoting depth in m. The distribution along the transect shown in Figure 1 of (b) Æ§ and (c) SA

[Figure]

**Fig. 3.** : The distribution along the transect shown in Figure 1 of (a) the concentrations of total Fe-binding ligand [Lt] and (b) concentrations of dissolved-Fe (DFe) and (c) a Æ§- SA diagram

[Figure]

**Fig. 4.** (a) The binding strength, log $K'_{Fe'L}$ $^{cond}$ and (b) complexation capacity, log $\alpha$FeÎĐL plotted in a Æ§-SA diagram. The color scale indicates the values of log $K'_{Fe'L}$ $^{cond}$ and log $\alpha$FeÎĐL.

[Figure]

**Fig. 5.** The distribution along the transect shown in Figure 1 of (a) excess ligand concentrations [LÎĐ], (b) Fluorescence, and (c) Nitrate.

[Figure]

**Fig. 6.** The distribution of Si*(a) and the ratio of [Nitrate]/DFe (b) along the transect shown in Figure 1.

---

## Author Response (AR1)

General comment:
This study investigated Fe-binding organic ligands distribution upper 600-m depth at 5 stations in the western Antarctic Peninsula (WAP). The research area covered the front and southern boundary of the Antarctic Circumpolar Current (ACC) as well as the zone influenced by the Coastal Current (CC) near the peninsula.
The results indicated that the organic ligands on the shelf were associated with icealgal exudates and melting sea-ice in surface water, and those in the deep shelf water were supplied via resuspension of shelf or sediments. The ligands concentration always exceeded dissolved Fe concentration, suggested that any additional Fe input can be stabilized in the dissolved form via organic complexation. Overall, this manuscript is well written and organized. But there are two points to be considered.

Reply:
Thank you for your kind words efforts to improve this manuscript. We have answered all the questions with the details listed below.

(1) The relationship between complexation capacity of the ligands and Fe distributions
The authors explained the relationship between complexation capacity of the ligands and Fe distributions in the beginning of section 4.2, but it was about the specific sample. How was the overall trend?
Reply:

In general, the complexation capacity was highest in mCDW, and lower in AASW compared to mCDW and uCDW. We have added text regarding the overall trend of complexation capacity in this study (lines 258 – 259).

"The largest log $K_{Fe'L}^{cond}$ and log $\alpha_{Fe'L}$ was measured in shelf waters, particularly in mCDW (mean log $\alpha_{Fe'L}$ =3.4±0.2, N=8; Figure 4b). The overall decreasing trend of log $K_{Fe'L}^{cond}$ and log $\alpha_{Fe'}$ was observed from mCDW, uCDW to AASW."

(2) The meaning of excess ligands in this study area, [L'] was always observed and additional Fe input was expected to be stabilized in the dissolved form. Although particulate Fe was not investigated in this study, it was expected that some portion of Fe might exist as particulate form in the WAP (Seyitmuhammedov, 2020). The co-existence of [L'] and particulate Fe sounds like a contradiction. How do the authors think about the contradiction? But I could not access the reference Seyitmuhammedov (2020) via online because it is Doctoral thesis; so I'm not sure whether Seyitmuhammedov (2020) researched the total dissolvable Fe during the same cruise to this study. If so, please explain brief results from Seyitmuhammedov (2020).
Reply:

In seawater, we can expect a steady state between dissolved and particulate Fe fractions. Dissolved Fe fractions consist predominantly of organic complexes. The particulate Fe fraction is formed by adsorption and aggregation processes. The particulate Fe fraction should be competing with the organically complexed dissolved Fe fraction, assuming that Fe is reversibly bound. The formation of particulate Fe fraction depends on pH, dissolved oxygen concentration and the concentration of dissolved Fe not bound by organic

ligands (Fe'). If the concentration of Fe' is above a threshold value, the particulate Fe will precipitate. This Fe' is thus governed by pH, adsorption sites on particles and competing strength of natural Fe-binding ligands. Competing strength of organic ligands is also called complexation capacity ($\alpha_{Fe'L}$), which depends on availability of ligand binding sites (concentration of 'free' ligands, [L']) and the conditional binding strength of ligands ($K^{cond}_{Fe'L}$,).

$\alpha_{Fe'L} = K^{cond}_{Fe'L} *[L']$

In summary, all species co-exist in a steady state, which depends pH, dissolved oxygen, $\alpha_{Fe'L}$ and adsorption sites on particles.

Briefly, the dissolved-Fe concentrations along the transect from shore to open ocean ranged from 0.08 – 4.88 nmol/L (Seyitmuhammedov et al., submitted). Relatively elevated concentrations of DFe (0.31 – 1.84 nmol/L) were observed in the surface layer at the shelf break and mid-shelf; and near to the seafloor on the shelf.

==

It is well recognized that the Fe speciation data in the ocean is important to understand Fe cycle in marine environment, the result and finding obtained in this study are valuable for future studies. Several minor comments are listed below.

Minor comments: Page 2, L38. CO2 "2" should be written in subscript.
Reply:
corrected.

Page 3, L72-79. Humic substances (HS) and HS-like substances. . . Complicated notation. Because this study did not investigate the HS and HS-like substances specifically, the authors can unify the terms and explain in this section.
Reply:
We have simplified the notation to become "humic or humic-like substances (HS)" in line 79.

Page 4, L114-115. Low density polyethylene bottle (LDPE, Nalgene). In general, GEOTRACES cookbook recommends fluorinated high density polyethylene bottle (FLPE) or Teflon bottles for the sampling of ligands in order to avoid the absorption to the bottle wall. Did the authors check the influence of the difference on the CLE-AdCSV?
Reply:

We have not specifically checked the influence of different bottle types (FLPE and LDPE) on CLE-AdCSV in **this** study. However, previous studies (Fischer et al., 2006; Gerringa et al., 2015; Jensen et al., 2020) have checked the adsorption of Fe on the bottle walls. Gerringa et al. (2015) reported that the DFe concentration in subsamples taken the ligand samples is about 13% lower compared to the DFe concentration in samples immediately acidified on board.

The adsorption of DFe in the bottle wall is not uniform and depends on the material of the bottles, whether they are new or old bottles, on the character of the ligands, on the saturation of the ligands. We conditioned the sample bottles by rinsing five times with the seawater sample before filling it.

Page 6, L137-. Section 2.3 Did the authors apply air purge method? Please add the information about the purging method.
Reply:

To improve the sensitivity of voltammetric measurement, air-blanketing and air-purging are recommended by Abualhaija et al. (2014) if samples are not already in equilibrium with air. The voltammetric cell in the BASi system (instrument used in this study) is open to the atmosphere, thus, we assumed that the sample in the cell is in equilibrium with air, therefore, we did not purge during the analysis. The sample in the voltammetric cell was stirred, *i.e.* as done by Buck et al. (2017).

Page 6, L158-159. . . .the product of [L'] and log K,. . . Probably the authors can eliminate "log" from the sentence.
Reply:
done.

Page 7, L164-165. The conditional stability constant of. . .. Did it mean that different calibration result from the original method (Abualhaija and van den Berg, 2014) was obtained?
Reply:
We have a different value than in Abualhaija and van den Berg (2014) since we used pH adjusted values of $\log\alpha_{inorg}$ = 10.4 for pH = 8.2 (Liu and Millero, 2002). These are slightly different from the commonly used $\log\alpha_{inorg}$ = 10. This $\log\alpha_{inorg}$ is used to transform the log $K^{cond}_{Fe^{3+}(SA)}$ in respect to $Fe^{3+}$, into the one with respect to Fe', log $K^{cond}_{Fe'(SA)}$. We obtained a conditional stability constant of SA log $K^{cond}_{Fe^{3+}(SA)}$ = 16.34 or log $K^{cond}_{Fe'(SA)}$ = 5.94, whereas (Abualhaija and van den Berg, 2014) obtained $K^{cond}_{Fe^{3+}(SA)}$ = 16.5 or log $K^{cond}_{Fe'(SA)}$ = 6.5. The results of our calibrations are reported by Gerringa et al. (submitted).

Page 8, Figure 2 (b)and(c). Please add the boundary line between mCDW and uCDW in Figures 2 (b) and (c).
Reply:
the approximate boundary line between mCDW and uCDW is added in figure 2b.

[Figure]

Page 9, Figure 3 (a), (b) and (c) Please add the titles for x-axis.
Reply:
Done.

Page 11, Figure 5 (b) and (c). Please add the data points in the Figures 5 (b) and (c), too.
Reply:
Done.

Page 12, L249- Section 4.1 Why there was the huge differences in [L'] distributions in deeper water between stations 70 and 72? Both stations are located in the shelf region but separated by a sill. It is very interesting. In the deeper waters at station 72, high Si* and low N/DFe values were observed. Is [L'] likely to have a relationship with Si* or N/DFe?
Reply:

We have done correlation tests for the correlations between [L'] and Si*; and between [L'] and N/DFe. There is no significant correlation between [L'] and N/DFe, whereas [L'] positively correlates with Si* ($R^2$=0.37; $F(1,25)$=14.85; $p$<0.01, coef. correlation r = 0.61). Since a negative value of Si* indicates Fe limitation, the positive correlation between [L'] and Si* indicates that the high [L'] in deeper water on the shelf was not related to Fe-limited conditions. Furthermore, the ratio of Mn:Fe at this station (Seyitmuhammedov et al., in review) indicates a lithogenic source of Fe at this location, thus it is possible that [L'] at station 72 was a result of sediment resuspension.

Page 13, L285 However, given. . .. I think the mixing process influenced on the distribution of phytoplankton as well as on those of Fe, L and nutrients. I think the ligand production rate by phytoplankton is different between species and their physiological status, too.
Reply:
We agree with the referee that different phytoplankton species have different rates of organic ligand production, and different species may also produce different type of organic ligands. We modified the existing paragraph with this information (lines 310 – 317).
"However, given the consistently low and constant distribution of [$L_t$] at the shelf break, it seems that mixing determines the distribution and net concentrations of ligands (Figure 3a) and the microbial species composition. Different microbial species have different rates of organic ligand production, and different microbial species may produce different type of organic ligands (Norman et al., 2015). The influence of mixing on ligands and microbial species composition is confirmed by the relatively constant distribution of DFe and macronutrients (i.e. nitrate; Figure 5c) at the same station, indicating that prominent mixing at the shelf break indeed is a major factor."

Page 16, L341-. Section 4.2. Are there any information about the phytoplankton species during this observation?
Reply:
we did not look at the phytoplankton species in particular, however, another group from the same cruise has examined the community composition in our study area (Joy-Warren et al., 2019). The study conducted by Joy-Warren et al. (2019) showed that Phaeocystis Antarctica dominated this region during our sampling period in Austral spring. We have added this information to the text (lines 380 – 383):

"A longer residence time has a positive feedback on the development of local primary productivity upon sea ice melting (Arrigo et al., 2017), supplying DFe to primary producers on the shelf, which were dominated by *Phaeocystis* Antarctica during our sampling period (Joy-Warren et al., 2019)."

Anonymous Referee #2

The study reports an investigation on the organic speciation of Fe along a natural gradient of the western Antarctic Peninsula. Although there were few stations investigated, the overall data can provide valuable information on the role of Fe and its ligands in a crucial area of the planet. The manuscript is generally well written and clear to follow.

Reply:

We thank the referee for these kind words, the thoughtful comments and efforts towards improving our manuscript.

I have some major remarks (mainly related to the presence and the quality of the data), along with some minor comments.

Major comments

1) I do not completely agree with the use of the reference Seyitmuhammedov 2020, being a PhD thesis not available. If it had been used just for a minor aspect, it would have been ok, but it is often cited, particularly for data that are present there but not presented in this manuscript. First of all, I think some additional detail for the DFe analysis (section 2.2) could be useful and I suggest to add them. However, the main problem is related to the values of labile particulate Fe and Mn (section 4.1), $\delta18O$ and dissolved and total-dissolvable Fe (section 4.2). In order to help readers, I think that they could be presented at least with ranges. Maybe it could have been smoother to publish those values before submitting this manuscript, to have a proper reference to cite.

Reply:

We have added additional detail for DFe analysis in section 2.2 (lines 139 - 152):

"The DFe analysis is described in detail by Seyitmuhammedov et al. (in review). In short, the DFe analysis was conducted using high-resolution inductively coupled plasma mass spectrometry (HR-ICP-MS) using a Thermo Fisher Element XR instrument at NIOZ, the Netherlands and using an Amtek Nu Attom instrument at the University of Otago, New Zealand. Samples were UV-oxidized and pre-concentrated using an automated seaFAST system (SC-4 DX seaFAST pico; ESI) equipped with Nobias-PA1 chelate resin. The quantification was done via standard additions. The recovery of the resin was ~100% and was verified in every analytical run by comparison between the slope of the seawater calibration curve and the eluent acid calibration curve after (Biller et al., 2012). Accuracy and reproducibility were monitored by regular measurements of the reference materials SAFe D1 and GEOTRACES South Pacific (GSP) seawater, and an in-house reference seawater sample, North Atlantic Deep Water (NADW). Results for DFe analyses of reference samples were 0.722 ± 0.008 nM (n = 3; NIOZ) and 0.729 ± 0.018 nM (n = 6; U. Otago) for SAFe D1 2013 (consensus value = 0.69 ± 0.04 nM) and 0.155 ± 0.045 nM (n = 13) for GSP 2019 consensus values. The average overall method blank (seaFAST and ICP-MS), determined by repeatedly measuring acidified ultrapure water in every analytical run as a sample, was 0.05 ± 0.02 nM (n = 21)."

The ratio of labile particulate Fe and Mn in the open ocean was 0.27 ± 0.49, and has been added in the text (lines 331 333).

"Additionally, the ratios of labile particulate Fe to labile particulate Mn (0.27 ± 0.49; Seyitmuhammedov et al. (in review) indicate that Fe has a biogenic origin in the offshore waters (Twining et al., 2004)."

the range of the $\delta^{18}O$ results from oxygen isotope analysis was added in lines 373 – 375.

"the results of oxygen isotope ($^{18}O$/$^{16}O$, conventionally reported into delta-notation as δ$^{18}O$; Seyitmuhammedov et al. (in review) analysis showed δ$^{18}O$ values ranged from -0.56 – 0.06 ‰. These values were used to estimate fractions of sea-ice meltwater (-1.9 – 1.1 %) and meteoric meltwater (precipitation and glacial;0.3 – 3.9 %)."

The range of DFe and total-dissolvable Fe in the study region during the sampling period was added in lines 397 – 401):

"the conditions along the WAP were not homogenous and elevated Fe ((up to 4.88 nM for DFe; (Seyitmuhammedov, in review)) concentrations northeast of our transect were observed in the upper 100 m, suggesting that some of the observed ligands might have been transported south-westerly with the CC."

The values of labile particulate Fe and Mn (section 4.1), δ18O and dissolved and total-dissolvable Fe will be available separately upon publication of Seyitmuhammedov et al. (in review). Similarly, the more detail procedure of DFe analysis will be available in Seyitmuhammedov et al. (in review).

2) I looked at the dataset presented in the reported link (https://doi.org/10.25850/nioz/7b.b.5) and I have some questions or remarks with the presented data and their use in the Results or Discussion sections.

2a. Fluorescence. What do negative values for fluorescence mean? Are they just a consequence of improper calibration or do they have another meaning? In addition, there are some fluorescence data missing (two depths for Station 70 and all the depths for Station 72), hence I wonder how the plots were drawn for Figure 5b. Please clarify.

Reply: Fluorescence data is obtained from a sensor attached to CTD rosette. The negative values only occur in the deeper than 100m where normally phytoplankton concentrations are really low. The calibration equation that is used to convert Volts to chl-*a* concentration in units of 'mg/m3' apparently creates negative values for the concentration when there is basically no chl-*a*. Thus, the negative values are basically 0 mg/m3. In our data, the lowest negative value is > -0.018, which is close to zero. The maximum fluorescence in the CTD file is about 2.43, and 0.018 is only a small percentage of the max. Finally, we have added the missing data and added the explanation above in the 'read me' text of table.

2b. DFe. Are data for St 90 40 e 100 m below the LOD? I ask that because that there is no standard deviation for those parameters, and also because the standard deviation of the blanks is reported as 0.02 nM (line 134), hence the LOD should be around 0.06 nM by using the 3σ method, which is higher than the values reported for those two samples (0.05 nM). If so, I think it should be clearly expressed, but in that case I wonder how the values could be plotted in Figure 3b (maybe as half the LOD?) and how the CLE-AdSV analyses were performed for those two samples, since they would need a value of DFe for the voltammetric titration. Please clarify this aspect and make the corrections if needed.

Reply: Yes, DFe at St90 at depth 40 and 100 was below the LOD and these samples are presented as the value of the LOD without an error. This is stated at the end of the paragraph 215. The LOD is 0.05 calculated from 3xSD of the blank (blank SD = 0.016 ; LOD is 0.054 ). However, we reported in 2 significant figures hence 0.05 instead of 0.06 nmol/L. We used the DFe concentration of 0.05 nM for these two samples.

2c. Silicate. Why data for Silicate are not reported in the table? Also, in line 314, to express the purpose of the Si* values, the authors comment that "a negative Si* indicates Fe limiting conditions", but in their dataset there are no negative values for Si*. Please explain better this point

Reply: we have added Silicate data to the table. Based on the cited literature, a negative value of Si* indicates Fe limiting conditions. In our dataset, there were no negative values for Si*, however, some values were close to 0 (e.g at the open ocean station). Our data indicates that although there is no Fe-limitation yet during our sampling period, Fe limitation could potentially occur later in the season. We have added the explanation above in the lines 333 – 340, as copied below:

"The lowest concentrations of DFe (<0.05 nM) were observed at St. D and E and were a result of both biological uptake and limited supply. This area most likely represent Fe-limited conditions as indicated by declining Si* (Si*= [Si] – [N]) values and high ratios of [nitrate]/DFe (Figures 6a and 6b). The value of Si* serves as a proxy for Fe limitation, where Fe stress leads to preferential drawdown of Si compared to N by diatoms in surface water (Takeda, 1998). A negative Si* indicates Fe limiting conditions, assuming that Si and N are required in a 1:1 ratio by diatoms (Brzezinski et al., 2002). In our dataset, although there were no negative values for Si*, some Si* values at the open ocean stations were close to 0. Our data indicates that although there is no Fe-limitation yet during our sampling period, Fe limitation could potentially develop later in the season."

Minor comments

- Line 38: correct CO2 ("2" in subscript).
  Reply:done

- Section 2.1: please define the material of the 0.2 µm filters used for filtration and the volume of the GO-FLO bottle. Although the conservation procedures are correct, I wonder why the samples for Fe-binding ligands and DFe were collected separately, instead of freezing just one bottle and take the aliquots for the two analyses from the same "container" in the lab (of course acidifying before DFe analysis).
  Reply: we have added the information on the material of the filters and added the volume of GO-FLO bottle (lines 1–23 - 124).
  "Seawater samples for DFe and Fe-binding ligands in this study were obtained using 12 L GO-FLO bottles attached to a Kevlar® wire. Seawater samples were filtered over 0.2 µm filters (cellulose acetate, Sartoban 300, Sartorius®) into pre-cleaned sample bottles inside a trace metal clean van."
  DFe samples were acidified immediately onboard to minimize the adsorption to the bottle wall. Based on previous studies (i.e Jensen et al. (2020)) and our experiences, the DFe concentration from ligand sample bottles are somewhat lower than the DFe concentration from immediately acidified samples due to precipitation in unacidified samples.

- Figure 1: I suggest using a darker yellow to indicate the Coastal Current.
  Reply: done.

- Section 2.2: please report the certified or informative values of SAFe D1 and GSP samples. In addition, report also the LOD of the procedure.
  Reply: we have included the certified values of SAFe D1 and GSP samples (lines 144 - 148). Also, the LOD of the procedure is added (lines 149 - 150).

- Line 153: in "CLE-CSV" there is an "Ad" missing before "CSV".
  Reply: Done.

- Line 156: the full stop at the end of the sentence is missing.
  Repy: done.

- Line 158: please close the parenthesis which was opened before "αFeÎDL".
  Reply: done.

- Line 160: the authors refer to αFeL, but I guess they meant αFeÎDL instead? ˇ
  Reply: corrected.

- Figure 2: please uniform the indication of "c." for the third figure, using the two parentheses consistently with (a) and (b). Also, in the caption, the "θ" in "σθ" should be in subscript. Finally, Absolute Salinity is reported with "A" in subscript or as plain SA in the text and in the Figures, please uniform in the whole manuscript.
  Reply: done.

- Line 189: I think there is some problem with the "" for Absolute Salinity. Did the authors mean "33.0 < SA < 33.7"?
  Reply: corrected.

- Figure 3: there is a reference missing (and an unclosed parenthesis) in "DFe, data from;". Also, what do the author mean when they say "with colors denoting depth the values of [Lt]"? I guess there's a "depth" in excess?
  Reply: corrected.

- Line 233: please remove the comma after [L'].
  Reply: done.

- Figure 5: why in some images the profiles are "smooth" (e.g. b) and in others are "rounded" (e.g. a and c)? Also, in Figure 5a there are only the profiles for the 5 stations, well separated, while for example in Figure 5c there are more. Why?
  Reply: Fe-binding ligand samples are only taken from 5 stations, therefore, in Figure 5a, where [L'] is presented, there are only the profiles from the 5 stations. For nutrient analyses (in this case, nitrate), samples were taken in a few more stations than ligand samples. Similarly, for Fluorescence data, this data is obtained from the sensor attached to CTD rosette, and fluorescence is recorded whenever the CTD is deployed to obtain seawater for many different analyses, thus we have higher resolution data for fluorescence.

- Line 264: I think the "that" is in excess?
  Reply: corrected.

- Line 284: since it is one value, it should be "maximum", while "maxima" is used for plurals (accordingly, correct also line 298 from "maximum" to "maxima" if it is referred to more than one).
  Reply: done.

- Line 298: unclosed parenthesis in "(St. 84 and 90; (Figure 5b)".
  Reply : done

- Figure 6: please insert the unit of measurement for Si*. Moreover, in the Figure there is "[Nitrate]/[DFe]" while in the caption there is "[Nitrate]/DFe", please uniform (DFe is presented without parentheses in the whole manuscript).
  Reply: done

- Line 325: please revise the "which commonly produced by" part, I do not think the sentence is fluid.
  Reply: the sentence is revised into two sentences.

- Line 367: "a phytoplankton blooms": it should be either "a phytoplankton bloom" or "phytoplankton blooms", please correct.
- Reply: corrected

- References are not well uniform in the use of the doi. Maybe there are also some parts missing (e.g. pages or article number for Arrigo 2008, Lam 2011, Lannuzel 2016, etc.).
Reply: we have checked the references and use the DOI uniformly in each reference. We also checked the journal volume and pages for all the references.

Anh Pham (Referee)
anhlpham@gatech.edu

In this manuscript, Ardiningsih et al. report measurements of the dissolved iron (Fe) and organic ligands concentrations, as well as the ligand conditional binding strength, in the upper 600-m water along five stations in the western Antarctic Peninsula (WAP) region of the Southern Ocean, where iron limits the ocean primary production. This region shows distinct features in the dissolved Fe pattern and in the hydrographic dynamics. It also contains biological hotspots in regions close to the shelf sea, where the marine ecosystem can be directly impacted by climate change. Thus, it is important to understand the mechanisms controlling the distribution of organic ligands and dissolved Fe in the WAP. The results of this manuscript suggest that ligands in the surface water of the shelf regions are the products of ice-algal exudates and sea-ice melting, while ligands in the deeper shelf water are produced from sediment-related processes. In the open ocean water of the WAP region, ligand productions are likely to be related to the sea-ice melting process and to the phytoplankton bloom. Mover, the authors show that the ligands concentration in the WAP always exceeds dissolved Fe concentration. This result suggests that any additional Fe input can be stabilized in the dissolved form, thereby being bioavailable for phytoplankton. Overall, this manuscript is well written and easy to follow. However, I would like to raise some questions and suggestions regarding the title, introduction, and discussion of the manuscript, which hopefully can be considered by the authors. I would be more than happy to backdown from these questions if the authors do not agree and provide good counter-arguments.

Thank you for the suggestions to improve the manuscript.

Major questions/comments

(1) First, the title (Sources of Fe-binding organic ligands in surface waters of the western Antarctic Peninsula) and the introduction of the manuscript give me the impression that this study will use a new method/technique other than CLE-AdCSV to identify the sources of organic ligands in the WAP region. In the Introduction, the authors wrote "The application of AdCSV gives the total concentration ([Lt]) and conditional binding strength of the dissolved organic ligands but does not provide information on the identity of ligands" (line 56-57) then stated that "the sources and identities of Fe-binding ligands are still largely unknown" (line 58=59). Thus, I was excited to see what (new) methods the authors would use to pinpoint the sources of organic ligands in the WAP region, which is a very important issue to address, and I think, have not been done before. However, at the end of the Introduction, the authors wrote (line 97-100): "In order to probe sources and distributions of Fe-binding ligands along a natural gradient of Fe, the CLE-AdCSV technique was used to quantify the total concentrations and conditional stability constants of Fe-binding ligands". To be honest, I was a bit disappointed and confused at this point. As an ocean biogeochemistry modeler who does not have a strong background on measurement techniques, I do not understand how the authors can probe the sources of ligands by using the CLE-AdCSV technique, which was stated before that cannot be used to provide information on the identity of ligands. It turns out that, if I understood correctly, the authors used CLE-AdCSV to measure the concentration and strength of organic ligands, then they based on other hydrographic and biogeochemical features, as well as previous studies, to hypothesize/argue about the potential origins of the measured ligands. While their arguments are valid, I think it is different from measurements that directly pinpoint to ligands' origin and identity. Thus, I would suggest the authors to modify the title and introduction such that they are not misleading and reflect correctly the problems that the manuscript directly address and the methods that the authors use to achieve this goal. Again, in my mind, this is a study that measures the ligands concentration and binding strength, then suggest their potential sources, not a study that directly identify the sources of ligands.

Reply: We agree with the referee that this study measures the concentration and binding strength of ligands. And then, the information is used to trace the potential sources of organic ligands. Therefore, we have revised some part of the introduction section, as listed below:
Tittle

- The title is changed into "Fe-binding ligands in coastal and frontal regions of the Western Antarctic Peninsula"

Introduction

- Lines 53 – 65: we modified the paragraph about the use of the Cle-AdCSV technique to measure the total concentration and conditional stability constant of ligands. We mentioned that the parameters obtained from the CLE-AdCSV analysis together with ancillary data can be used to infer the potential source of organic ligands in seawater.

"Different ligand types exist with characteristics (*i.e* binding strength) that are probably related to their origin. The characteristics of organic ligands can be measured by the competition against well-characterized artificial ligands with known stability constants. Analysis is done using an electrochemical technique, competitive ligand exchange (CLE) - adsorptive cathodic stripping voltammetry (AdCSV). The application of AdCSV gives the total concentration ($[L_t]$) and conditional binding strength ($K_{Fe'L}^{cond}$) of the dissolved organic ligands but does not provide information on the identity of ligands. Even though the exact compositions and origins of Fe-binding ligands are still largely unknown, $[L_t]$ and $K_{Fe'L}^{cond}$ obtained by CLE-AdCSV, together with ancillary data, can be used to infer potential sources of these organic ligands. The organic ligands in seawater either have a terrestrial or marine source. The terrestrial-sourced ligands are supplied from lithogenic and pedogenic inputs within the boundary region between land and sea (i.e coastal seas and estuaries) (Buck and Bruland, 2007; Batchelli et al., 2010; Kondo et al., 2007; Buck et al., 2007; Bundy et al., 2015; Gerringa et al., 2007; Laglera and van den Berg, 2009). The marine organic ligands come from *in situ* biological activities, being either actively produced or passively generated through microbial activity."

- Lines 102 – 106: we have revised the sentences to reflect the goals of this manuscript.

"In this study, surface waters were sampled in a region of mixing between shelf-influenced waters and HNLC waters in the Bellingshausen Sea along the WAP. The CLE-AdCSV technique was used to quantify the total concentrations and conditional stability constants of Fe-binding ligands. These parameters were used to examine the distribution and identify the potential sources of organic ligands from ice covered shelf waters to the open ocean of the Antarctic Zone."

(2) Second, in section 4.2 of the manuscript, the authors discussed at length on how a high complexation capacity of ligand and ice-melting processes can control the ocean primary productivity in the WAP region. They also discussed on the potential impact of global warming on Fe chemistry and ligand, and stated that (line 387-390): "Overall, the continued sea-ice melt and glacial retreat can be expected to increase the supply of Fe (Lannuzel et al., 2016), other micronutrients (Co, Mn, etc.), and Fe-binding ligands (Lin and Twining, 2012), but the consequences for their complexation capacity and overall bio-availability of Fe remain elusive." But what about the impact of ligand production from ice-algal exudates, sediments, and phytoplankton bloom? How will these processes change in the future under the impact of global warming? Is the ligand production from ice-algal exudates going to increase or decrease with sea-ice melting? Since these are major processes in producing ligands, I would love to see more discussion on them.
Reply: Environmental alteration due to global warming influences the dynamics of marine ecosystems (as mentioned in lines 380 – 385), and thus the production of ligands via many different processes (i.e production from ice-algal exudates, sediments, and phytoplankton blooms). We have implicitly mentioned the potential impact of ongoing environmental changes on ligand production associated with phytoplankton blooms (lines 389 - 394). We have added discussion

related to the potential release of organic ligands from sediment entrapped with sea-ice (line 397) and organic ligands associated with microbial excretion (lines 398 – 400).

"Changes in planktonic community composition affect net primary production and overall carbon drawdown, which lead to further alteration of the food web and carbon cycling (Alderkamp et al., 2012; Arrigo et al., 1999; Joy-Warren et al., 2019; Schofield et al., 2017). These and other ongoing changes in the food web will also affect production of dissolved organic carbon (DOC) and thus ligands as they form a fraction of the DOC pool (Gledhill et al., 2012; Whitby et al., 2020). Generally, one expects that increased DOC production would lead to more ligands, but the binding strength depends on which molecules are formed (Gledhill and Buck, 2012; Hassler et al., 2017). Additionally, intensified light exposure alters log $K_{Fe'L}^{cond}$ by photo-oxidative processes, possibly reducing the complexation capacity and binding strength for Fe (Barbeau et al., 2001; Mopper et al., 2015; Powell et al., 2003) as well as the bioavailability (Hassler et al., 2020). Furthermore, complexation capacity is affected by pH, implying that ongoing ocean acidification also influences the speciation of Fe (Ye et al., 2020). The melting of black sea-ice entrapped with sediment potentially releases organic ligands (Genovese et al., 2018). Organic ligands from microbial excretions are expected to be abundant on the base of sea ice (Norman et al., 2015), although the fluctuation rate (decrease or increase) under on-going changes cannot be confirmed without further experiment."

Minor comments:
- Line 16: Our results indicate that organic ligands in "the" surface water. . . Corrected.
- Line 17: Organic ligands in "the" deeper shelf water. Corrected.
- Line 41: Should the reference here be Henley et al., 2019? Corrected
- Line 69-70: covered by sea-ice (remove with) Done.
- Line 85-86: such as glacial meltwater, sediments", and" upwelling. Corrected.
- Line 171: hydrographic features of the WAP "was" described elsewhere. Corrected.
- Line 173: Two distinct horizontal currents exist in the study area: The Coastal Current
- (CC) and the Antarctic Circumpolar (replace, by: ) Corrected.
- Figure 3 caption: remove depth in colors denoting depth the values. . . Done.
- Line 323-326: Maybe revise this sentence to make it shorter and clearer: "Mopper et al. (2015) suggested that the absorption of solar radiation by chromophoric dissolved organic matter as part of the ligand pool which commonly produced by sea ice algae (Norman et al., 2011), leads to the photochemical transformation of these compounds. The sentence is divided into two sentences.

Jonathan Lauderdale (Referee)
jml1@mit.edu

In this manuscript, Ardiningsih and coauthors present biogeochemical observations from a cruise transect from the western Antarctic Peninsula offshore into the Southern Ocean. This region is particularly susceptible to changes in climate. Three different hydrographic regions are identified, influenced by watermass type and sea ice cover that are suggested to host different distributions and characteristics of dissolved iron and organic iron-chelating ligands: firstly, surface "winter waters" near the coast and on the continental shelf are strongly influenced by sea ice cover with organic ligand production associated with ice-associated algae and iron supply from glacier melt; secondly, upwelling deep waters on the continental slope are initially low in iron and ligands, but concentrations increase as a result of sediment-water interactions and resuspension; and thirdly, in offshore waters of the Antarctic Zone influenced by seasonal sea ice melt, phytoplankton blooms deplete nutrients and iron, while actively or passively producing organic ligands. I thought this paper was logically organized and engagingly written, and I think the authors do a good job of balancing the fact that the CLE-AdCSV methodology gives information about how much and how strong the ligand(s) present are but not what compounds, with some well-reasoned evidence-based assumptions. I was particularly interested to read about the tube-dwelling sea ice diatoms, and the interaction between sediments and upwelling circumpolar deep waters. However, I thought the presentation of results was not optimal. I did find the TS diagrams very informative – nevertheless, all the figures had some issues:

Reply: Thank you for the suggestions to improve the presentation of the results. We have revised the figures as suggested with details below.

1.) The yellow coastal current arrow blends in to the pale blue bathymetry in Figure 1 (also the "CC" triangle in Fig.2). I would have liked to see actual surface current data (maybe LADCP or something?) at the station locations, and how they compare to the broad-brush arrows used to mark the ACC jets.
Reply: We have changed the color into darker yellow and changed the color scheme of the map. Arrigo et al. (2017) has described the velocity profiles (section 2.7 and Figure 6), which are calculated from Acoustic Doppler Current Profiler (ADCP) data and CTD measurements. They found a current in the direction of ACC flow (eastward) about 100 – 250 km from the shelf break. the broad-bush arrows in Fig.1 were only an approximation of the ACC flow. The transect in this study is the sampling line 300 in Arrigo et al. (2017), where SACCF and SB were 100-150km apart. We have referred to Arrigo et al. (2017) in the caption of Figure 1 for the details of ACC flow

Figures 2-6 use a rainbow color palette that is firstly not universally accessible to those with color vision deficiency, and secondly creates a perceptually non-uniform color space that can create misleading gradients in continuous data such as temperature and salinity, as well as dissolved iron and ligand concentrations (e.g. https://blogs.egu.eu/divisions/gd/2017/08/23/the-rainbow-colourmap/ or https://www.nature.com/articles/s41467-020-19160-7).
Reply: Thank you for the additional information in the links and your thoughtful suggestions. We have changed the color scheme of the figures to the viridis color palette as this should be universally accessible to those with color vision deficiency.

2.) The use of "Ocean Data View" section plots is an issue for me, both because of the colormap, but also because of the variable size of each colored "blob" for bottle data. In Fig.5a for example, station 84 has wide circles while station 90 has narrower circles (and 96 has even narrower), yet it is not true that station 84's concentrations are applicable on a larger spatial scale than station 90's (or even station 96's) – the fluorescence data in Fig.5b clearly has much shorter spatial variability (from more stations/continuous CTD cast data?) that is similar at all stations. ODV's "patchiness" really emphasizes the sparsity of the ligand and iron measurements here, which is a shame to detract

from a precious dataset that clearly has interesting information within it – perhaps horizontally-stacked line profiles might work better than contours? As I said, in general I found the TS plots were more interesting and useful. Finally, a minor point is that I found the station ordering to be a bit confusing because they are presented in chronological order, while the manuscript suggests an order that is more geographically orientated. I wonder whether the authors could use an alphabetical scheme for this paper with a key in Fig.1 relating the letters back to station number for posterity (so, the stations would be A (72) to E (90) according to distance from the coast)?

Reply:

Thank you for the thoughtful suggestions to improve the presentation of the results. We have changed the station number into alphabetical scheme according to distance from coast. The section figure in ODV is best to describe data along a transect, thus we would like to keep the section figures. However, in order to avoid ODV's patchiness, we used color dots to present the data from Figure 3 to 6. Below is an example of the revised figures.

Figure 1.

[Figure]

Figure 2

[Figure]

**Figure 3**

[Figure]

[Figure]

References

Abualhaija, M.M. and van den Berg, C.M.G., 2014. Chemical speciation of iron in seawater using catalytic cathodic stripping voltammetry with ligand competition against salicylaldoxime. Marine Chemistry, 164: 60-74.

Alderkamp, A.-C., Kulk, G., Buma, A.G.J., Visser, R.J.W., Van Dijken, G.L., Mills, M.M. and Arrigo, K.R., 2012. The effect of iron limitation on the photophysiology of phaeocystis antarctica (prymnesiophyceae) and fragilariopsis cylindrus (bacillariophyceae) under dynamic irradiance. Journal of Phycology, 48(1): 45-59.

Arrigo, K.R., Robinson, D.H., Worthen, D.L., Dunbar, R.B., DiTullio, G.R., VanWoert, M. and Lizotte, M.P., 1999. Phytoplankton Community Structure and the Drawdown of Nutrients and CO2 in the Southern Ocean. Science, 283(5400): 365-367.

Arrigo, K.R., van Dijken, G.L., Alderkamp, A.-C., Erickson, Z.K., Lewis, K.M., Lowry, K.E., Joy-Warren, H.L., Middag, R., Nash-Arrigo, J.E., Selz, V. and van de Poll, W., 2017. Early Spring Phytoplankton Dynamics in the Western Antarctic Peninsula. Journal of Geophysical Research: Oceans, 122(12): 9350-9369.

Barbeau, K., Rue, E.L., Bruland, K.W. and Butler, A., 2001. Photochemical cycling of iron in the surface ocean mediated by microbial iron(iii)-binding ligands. Nature, 413(6854): 409-413.

Biller, D.V. and Bruland, K.W., 2012. Analysis of Mn, Fe, Co, Ni, Cu, Zn, Cd, and Pb in seawater using the Nobias-chelate PA1 resin and magnetic sector inductively coupled plasma mass spectrometry (ICP-MS). Marine Chemistry, 130: 12-20.

Brzezinski, M.A., Pride, C.J., Franck, V.M., Sigman, D.M., Sarmiento, J.L., Matsumoto, K., Gruber, N., Rau, G.H. and Coale, K.H., 2002. A switch from Si(OH)4 to NO3− depletion in the glacial Southern Ocean. Geophysical Research Letters, 29(12): 5-1-5-4.

Buck, K.N., Sedwick, P.N., Sohst, B. and Carlson, C.A., 2017. Organic complexation of iron in the eastern tropical South Pacific: Results from US GEOTRACES Eastern Pacific Zonal Transect (GEOTRACES cruise GP16). Marine Chemistry.

Fischer, A.C., Wolterbeek, H.T., Kroon, J.J., Gerringa, L.J.A., Timmermans, K.R., van Elteren, J.T. and Teunissen, T., 2006. On the use of iron radio-isotopes to study iron speciation kinetics in seawater: A column separation and off-line counting approach. Science of The Total Environment, 362(1): 242-258.

Genovese, C., Grotti, M., Pittaluga, J., Ardini, F., Janssens, J., Wuttig, K., Moreau, S. and Lannuzel, D., 2018. Influence of organic complexation on dissolved iron distribution in East Antarctic pack ice. Marine Chemistry, 203: 28-37.

Gerringa, L.J.A., Gledhill, M., Ardiningsih, I., Muntjewerf, N. and Laglera, L., submitted. Comparing CLE-AdCSV applications using SA and TAC to determine the Fe binding characteristics of model ligands in seawater Frontiers in Marine Science.

Gerringa, L.J.A., Laan, P., van Dijken, G.L., van Haren, H., De Baar, H.J.W., Arrigo, K.R. and Alderkamp, A.C., 2015. Sources of iron in the Ross Sea Polynya in early summer. Marine Chemistry, 177, Part 3: 447-459.

Gledhill, M. and Buck, K., 2012. The organic complexation of iron in the marine environment: A review. Frontiers in Microbiology, 3(69).

Hassler, C., Cabanes, D., Blanco-Ameijeiras, S., Sander, S.G. and Benner, R., 2020. Importance of refractory ligands and their photodegradation for iron oceanic inventories and cycling. Marine and Freshwater Research, 71(3): 311-320.

Hassler, C.S., van den Berg, C.M.G. and Boyd, P.W., 2017. Toward a Regional Classification to Provide a More Inclusive Examination of the Ocean Biogeochemistry of Iron-Binding Ligands. Frontiers in Marine Science, 4(19).

Jensen, L.T., Wyatt, N.J., Landing, W.M. and Fitzsimmons, J.N., 2020. Assessment of the stability, sorption, and exchangeability of marine dissolved and colloidal metals. Marine Chemistry, 220: 103754.

Joy-Warren, H.L., van Dijken, G.L., Alderkamp, A.-C., Leventer, A., Lewis, K.M., Selz, V., Lowry, K.E., van de Poll, W. and Arrigo, K.R., 2019. Light Is the Primary Driver of Early Season Phytoplankton Production Along the Western Antarctic Peninsula. Journal of Geophysical Research: Oceans, 124(11): 7375-7399.

Mopper, K., Kieber, D.J. and Stubbins, A., 2015. Chapter 8 - Marine Photochemistry of Organic Matter: Processes and Impacts. In: D.A. Hansell and C.A. Carlson (Editors), Biogeochemistry of Marine Dissolved Organic Matter (Second Edition). Academic Press, Boston, pp. 389-450.

Norman, L., Worms, I.A.M., Angles, E., Bowie, A.R., Nichols, C.M., Ninh Pham, A., Slaveykova, V.I., Townsend, A.T., David Waite, T. and Hassler, C.S., 2015. The role of bacterial and algal exopolymeric substances in iron chemistry. Marine Chemistry, 173: 148-161.

Powell, R.T. and Wilson-Finelli, A., 2003. Photochemical degradation of organic iron complexing ligands in seawater. Aquatic Sciences, 65(4): 367-374.

Schofield, O., Saba, G., Coleman, K., Carvalho, F., Couto, N., Ducklow, H., Finkel, Z., Irwin, A., Kahl, A., Miles, T., Montes-Hugo, M., Stammerjohn, S. and Waite, N., 2017. Decadal variability in coastal phytoplankton community composition in a changing West Antarctic Peninsula. Deep Sea Research Part I: Oceanographic Research Papers, 124: 42-54.

Seyitmuhammedov, K., Stirling, C.H., Reid, M.R., van Hale, R., Laan, P., Arrigo, K.R., van Dijken, G., Alderkamp, A.-C. and Middag, R., in review. The distribution of Fe across the shelf of the Western Antarctic Peninsula at the start of the phytoplankton growing season. Marine Chemistry.

Takeda, S., 1998. Influence of iron availability on nutrient consumption ratio of diatoms in oceanic waters. Nature, 393(6687): 774-777.

Twining, B.S., Baines, S.B. and Fisher, N.S., 2004. Element stoichiometries of individual plankton cells collected during the Southern Ocean Iron Experiment (SOFeX). Limnology and Oceanography, 49(6): 2115-2128.

Whitby, H., Planquette, H., Cassar, N., Bucciarelli, E., Osburn, C.L., Janssen, D.J., Cullen, J.T., González, A.G., Völker, C. and Sarthou, G., 2020. A call for refining the role of humic-like substances in the oceanic iron cycle. Scientific reports, 10(1): 6144-6144.

Ye, Y., Völker, C. and Gledhill, M., 2020. Exploring the Iron-Binding Potential of the Ocean Using a Combined pH and DOC Parameterization. Global Biogeochemical Cycles, 34(6): e2019GB006425.